# Proteomic patterns associated with response to breast cancer neoadjuvant treatment

Anjana Shenoy[1] [ID], Nishanth Belugali Nataraj[2] [ID], Gili Perry[3], Fabricio Loayza Puch[4], Remco Nagel[4], Irina Marin[5], Nora Balint[5], Noa Bossel[2], Anya Pavlovsky[5], Iris Barshack[5], Bella Kaufman[6], Reuven Agami[4] [ID], Yosef Yarden[2] [ID], Maya Dadiani[3] & Tamar Geiger[1],* [ID]

## Abstract

Tumor relapse as a consequence of chemotherapy resistance is a major clinical challenge in advanced stage breast tumors. To identify processes associated with poor clinical outcome, we took a mass spectrometry-based proteomic approach and analyzed a breast cancer cohort of 113 formalin-fixed paraffin-embedded samples. Proteomic profiling of matched tumors before and after chemotherapy, and tumor-adjacent normal tissue, all from the same patients, allowed us to define eight patterns of protein level changes, two of which correlate to better chemotherapy response. Supervised analysis identified two proteins of proline biosynthesis pathway, PYCR1 and ALDH18A1, that were significantly associated with resistance to treatment based on pattern dominance. Weighted gene correlation network analysis of post-treatment samples revealed that these proteins are associated with tumor relapse and affect patient survival. Functional analysis showed that knockdown of PYCR1 reduced invasion and migration capabilities of breast cancer cell lines. PYCR1 knockout significantly reduced tumor burden and increased drug sensitivity of orthotopically injected ER-positive tumor *in vivo*, thus emphasizing the role of PYCR1 in resistance to chemotherapy.

**Keywords** breast cancer; chemotherapy; mass spectrometry; proline biosynthesis; proteomics
**Subject Categories** Cancer; Pharmacology & Drug Discovery; Proteomics
**Mol Syst Biol. (2020) 16: e9443**

## Introduction

Breast cancer is the leading cause of female cancer-related death. While survival rates are very high when diagnosed early, treatment of large tumors and metastatic disease is more challenging.

Treatment decisions are based on the breast cancer subtype, and the tumor stage at diagnosis. Hormone-positive subtypes that express the estrogen receptor (ER) and/or the progesterone receptor (PR) can be treated with hormonal therapy (e.g., tamoxifen). Her2-positive subtype can be targeted by herceptin. The main available option to treat triple-negative breast cancer (TNBC), which do not express any of these receptors, is chemotherapy and surgery (Guarneri & Conte, 2009). These clinical subtypes are linked to molecular classification based on gene expression analysis. Four main subtypes include the following: luminal A ($ER^+$ and/or $PR^+$, $HER2^-$, $Ki67^{low}$), luminal B ($ER^+$ and/or $PR^+$, $HER2^+$, $Ki67^{high}$), basal-like ($ER^-$, $PR^-$, $HER2^-$), and Her2 overexpressing ($ER^-$, $PR^-$, and $HER2^+$; Perou *et al*, 2000; Sorlie *et al*, 2001, 2003). The subtypes were reproducibly identified in several studies and also reflect clinical outcomes, with basal-like subtype having the worst, and luminal A subtype having the best prognosis (Hodgkinson *et al*, 2010; The Cancer Genome Atlas Network, 2012).

Beyond subtype-specific treatment decisions, tumor TNM staging dictates the therapeutic approach. Locally advanced invasive tumors and tumors that are difficult to operate are considered for neoadjuvant treatment (NAT), which is given before surgery (Saleh *et al*, 2014). Although NAT is administered to shrink large tumors, it is increasingly used to obtain prognostic information about tumor chemo-sensitivity by evaluation of pathological complete response (pCR). pCR is defined as the complete eradication of malignant disease in the breast and related axillary lymph nodes on completion of NAT and is often associated with improved survival rates (Perou *et al*, 2000; Goldstein *et al*, 2007; Cortazar *et al*, 2014). The lowest pCR rate is found among luminal tumors (6.4–22%) and highest among Her2 overexpressing and triple-negative subtypes (27–32%) (Cortazar *et al*, 2014; Broglio *et al*, 2016). The extent of residual disease in partial responders after NAT is variable and is an important prognostic factor to predict risk of relapse and overall survival (OS; Denkert *et al*, 2011; Symmans *et al*, 2017). Consequently, there is a major clinical need to understand the underlying resistance mechanisms that eventually lead to tumor recurrence.

1  Sackler Faculty of Medicine, Tel Aviv University, Tel Aviv, Israel
2  Weizmann Institute of Science, Rehovot, Israel
3  Sheba Medical Center, Cancer Research Center, Tel-Hashomer, Israel
4  Netherlands Cancer Institute, Amsterdam, Netherlands
5  Sheba Medical Center, Pathology Institute, Tel-Hashomer, Israel
6  Sheba Medical Center, Oncology Institute, Tel-Hashomer, Israel
   *Corresponding author. Tel: +972 52 8611611; E-mail: geiger@tauex.tau.ac.il

Previous breast cancer studies analyzed tumor tissues utilizing cDNA microarray and sequencing and identified markers of drug resistance such as CSNK2B, DDB1, ABL, PRKDC, and DUSP4 that were differentially expressed between responders and non-responders to NAT (Chang *et al*, 2003; Balko *et al*, 2012). Using reverse-phase protein arrays of 76 proteins, Sohn *et al* identified AKT, IGFBP2, LKB1, S6, and Stathmin as predictors of recurrence-free survival (RFS) in triple-negative breast cancer (TNBC; Sohn *et al*, 2013). These studies highlighted the importance of cancer signaling in eliciting resistance. MS-based clinical proteomics of breast cancer has focused in recent years on cancer classification, showing protein networks associated with each subtype, with driver mutations and was able to challenge the RNA-based classification (Mertins *et al*, 2016; Tyanova *et al*, 2016a; Yanovich *et al*, 2018). Recently, proteogenomic analysis of breast cancer treatment response showed proteins associated with Herceptin resistance in a small patient cohort (Satpathy *et al*, 2020). We hypothesized that an untargeted, proteomic approach has the potential to unravel novel pathways of neoadjuvant chemotherapy response.

In this study, we performed LC-MS/MS proteomic analysis of formalin-fixed paraffin-embedded (FFPE) tissues to understand response to NAT in breast cancer. We analyzed matched tumors before and after NAT, with matched tumor-adjacent normal samples from 35 patients with partial response to chemotherapy. We measured the response to chemotherapy using the previously described Miller & Payne pathological response score (M&P score; Ogston *et al*, 2003), and examined the factors associated with tumor recurrence. The longitudinal analysis highlighted protein expression patterns associated with pathological response and recurrence, revealing two proteins in the proline biosynthesis pathway. Finally, functional analysis *in vivo* showed that abundance of PYCR1, a mitochondrial metabolic protein, was associated with drug resistance to chemotherapy, thus stressing the role of this protein in breast cancer.

## Results

### Proteomic analysis of response to neoadjuvant treatment

To identify the proteomic alterations that result in poor response to NAT, we assembled 108 FFPE tissue samples from a cohort of 35 partial responders to NAT. One patient had bilateral invasive cancer, and samples from both breasts were analyzed separately. We obtained matched pre-treatment biopsy, post-treatment residual carcinoma, and tumor-adjacent normal samples from each of the 35 patients. To ensure that the tumor-adjacent normal samples are suitable as controls, we compared them to breast ducts from five breast reduction surgeries of healthy women bringing us to 113 samples (Appendix Fig S1A). The tumor specimens were pathologically classified as triple-positive (ER, PR, and HER2 positive; $n = 9$), triple-negative (ER, PR, and HER2 negative; $n = 5$), Her2 overexpressing ($n = 1$), and hormone-positive (ER/PR positive; $n = 21$) subtypes. The average follow-up time in the cohort was five years, and seven patients showed relapse post-treatment (Fig 1A, Dataset EV1). For every patient in the cohort, we compared the reduction in tumor cellularity before and after treatment and calculated the Miller & Payne (M&P) pathological response score (MP1: no reduction in tumor cellularity/poor

responders, $n = 6$; MP2: minimum reduction in cellularity/partial responders, $n = 13$; MP3: partial reduction in cellularity/better responders, $n = 16$; MP4: significant reduction in cellularity/better responders, $n = 1$, grouped with MP3 as a better responder). A higher M&P score (MP5) represents full responders with more than 90% reduction in cellularity and was not included in this study since these have no post-treatment specimens. Patients assigned to the three M&P groups showed significantly different relapse rates, and relapse-free survival time (RFS; Fig 1B and C).

We performed LC-MS/MS-based proteomic analysis on each sample, with a common super-SILAC mix as a heavy-labeled reference sample for accurate quantification. Due to limited sample amount (from biopsies and from post-treatment samples), we reached a partial proteome coverage of 7,600 identified and 7,180 quantified proteins in total, with 3,200–3,900 proteins in each sample group (Appendix Fig S1B, Dataset EV2).

Principal component analysis (PCA) showed a clear separation between tumor-adjacent normal samples and tumor samples (pre- and post-treatment) in the first two components (Appendix Fig S1C). Healthy samples from breast reduction surgery (non-transformed normal samples) and the tumor-adjacent normal samples were not separated, and had no significant differences between them (Student's *t*-test FDR < 5%), thus confirming the longitudinal analysis of matched adjacent normal samples as a normal breast reference.

In agreement with the PCA analysis, hierarchical clustering of sample correlations separated between normal and cancer samples (Fig 1D). Notably, the matched pre-treatment and post-treatment tumor samples from 19 patients co-clustered significantly ($P < 0.05$). Interestingly, we found that pre- and post-treatment co-clustering, representing higher proteome correlations and reduced treatment effect, also showed significantly higher relapse ($P = 0.051$) and poor pathological response ($P = 0.035$) compared to patients that did not show co-clustering of tumor samples (Fig 1E and F). Spearman rank correlations of all samples ranged between 0.18 and 0.82 (Appendix Fig S1D). Average correlation between matched pre- and post-treatment samples was 0.58, and the correlations with matched tumor-adjacent normal were markedly lower (post-treatment: tumor-adjacent normal ($R = 0.43$) and pre-treatment: tumor-adjacent normal ($R = 0.39$; Appendix Fig S1E).

### Pattern analysis links pattern 3 to poor survival

Pathological response is reflected in the change that occurs between two tumor states (pre- and post-treatment), rather than a snapshot of one of them. Therefore, in order to identify resistance mechanisms from the protein abundance data, we took advantage of the matched nature of our cohort. For each patient, we divided the proteins into eight profiles of protein level changes between normal, pre-treatment tumors, and post-treatment tumor samples (Dadiani *et al*, 2016). For example, pattern 1 consists of all the proteins upregulated in the pre-treatment sample compared to the tumor-adjacent normal sample and downregulated post-treatment (Fig 2A). We first performed the analysis using three paired Student's *t*-tests, each between two groups with an FDR cutoff of 5% across all patients (henceforth referred to as global pattern analysis). Significantly changing proteins (904 proteins) followed patterns 1, 2, 3, 4, and 6, across the 36 matched samples, and no

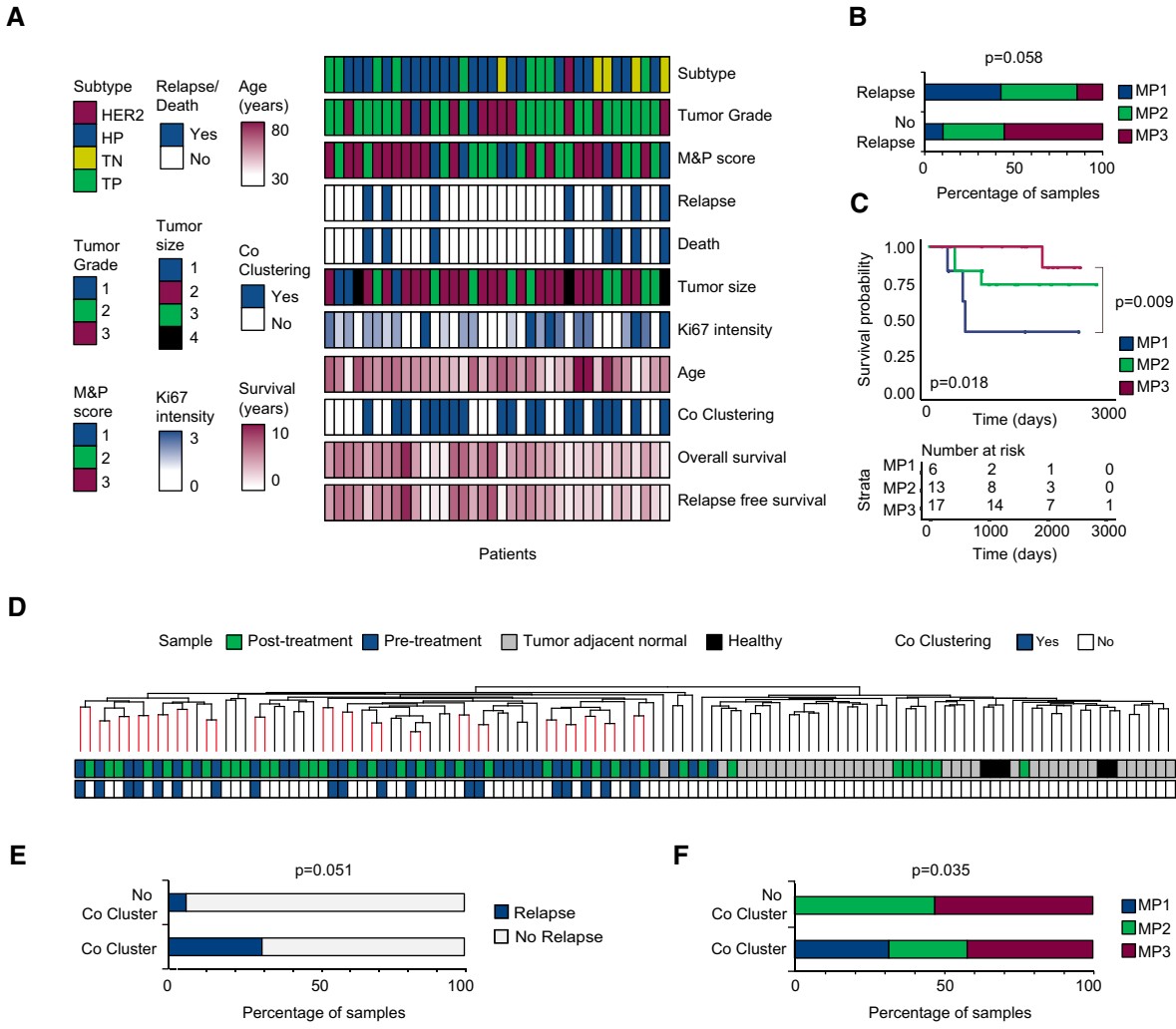

**Figure 1. Unsupervised analysis of proteomics of neoadjuvant treatment.**

A  Clinical parameters of 35 patients. Subtype at diagnosis (HP: ER+ and/or PR+, TP: ER+ and/or PR+, HER2+, TN: ER−, PR−, HER2− and Her2 overexpressing: ER−, PR−, and HER2+, H: healthy breast ducts, N: tumor adjacent normal); tumor grade at diagnosis (1, 2, or 3); Miller & Payne score(1: no reduction in tumor cellularity, 2: minimum reduction in cellularity, 3: partial reduction in cellularity); relapse (yes/no); death (yes/no); Ki67 intensity measured by immunohistochemistry at diagnosis (1–3); tumor size at diagnosis (1: < 2 cm, 2:2–5 cm, 3:> 5 cm, 4 indicates that tumor has grown into chest wall or skin); age (years); co-clustering of matched pre- and post-treatment samples after hierarchical clustering (yes/no, $P < 0.05$); relapse-free survival time (years); overall survival time (years).

B  Association between patient pathological response and tumor relapse in our cohort. Significance was determined using chi-square test.

C  Kaplan–Meier survival curve for Miller and Payne pathological response score. Global log rank $P$ value and corrected $P$ value for pairwise comparison are indicated.

D  Dendrogram colors indicate co-clustering of matched pre- and post-treatment tumor samples coming from the same patient.

E  Association between co-clustering of matched pre- and post-treatment tumor samples with relapse. Significance was determined using chi-square test.

F  Association between co-clustering of matched pre- and post-treatment tumor samples with M&P score. Significance was determined using chi-square test.

significant proteins followed patterns 5, 7, and 8 (Appendix Fig S1F, Dataset EV3). We postulated that proteins dominated by global patterns 1 and 2, which revert to normal levels upon treatment, are associated with good prognosis, while proteins that mostly follow patterns 3, 4, 7, and 8, which remain significantly different from the tumor-adjacent normal even after treatment, may be associated with poor response. Pattern 3, which includes proteins significantly upregulated in cancer, that are not affected by treatment, was the most dominant pattern, with 736 proteins. Centrality analysis of pattern 3 protein network highlighted several oncogenes among the top 10% most central proteins, including AKT, MTOR, and STAT1,

suggesting a role for these proteins in forming specific patterns of protein level changes and treatment resistance (Fig 2B and C).

To directly associate the pattern of protein changes to treatment response in individual patients, we defined a patient-wise protein pattern for every protein in each patient, using a fold change cutoff of 1.5 for protein level changes between matched samples (Dataset EV4). Percentage of each patient-wise pattern in the different patients shows dominance of patterns 1–4 in all patients (Fig 2D). In support of our hypothesis, percentage of pattern 3 proteins in patients was associated with shorter relapse-free survival ($R = −0.45$, $q = 0.03$; Fig 2E).

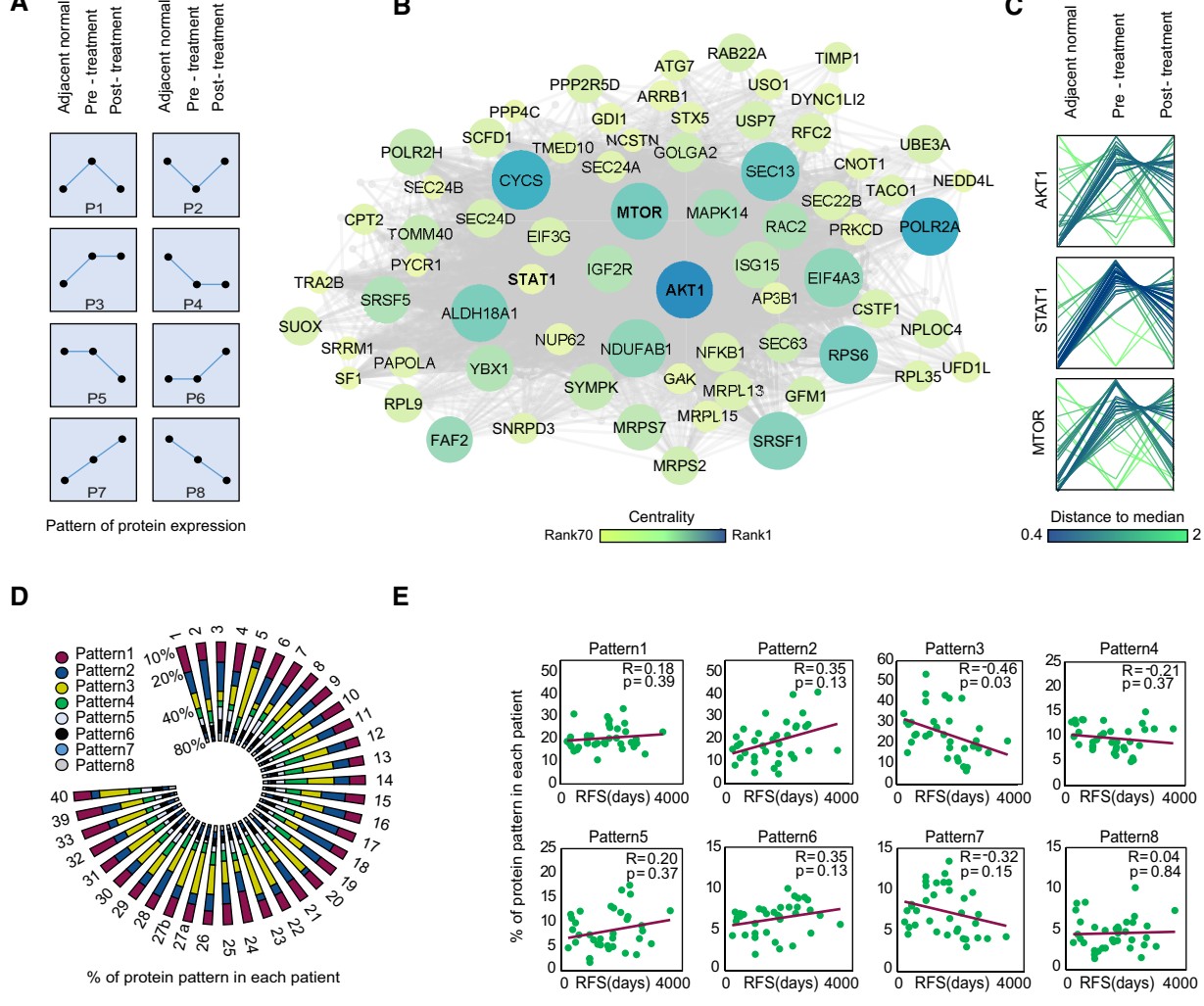

**Figure 2. Pattern analysis associates proteins with patient prognosis.**

A   Eight patterns of protein level changes between the three clinical groups: adjacent normal, pre-treatment tumor, and post-treatment tumor.

B   Pattern 3 protein network based on global pattern analysis: Node size and color are based on betweenness centrality score of each node in the network. Top 10% of the most central nodes are indicated

C   Profile plot shows z-scored protein pattern of selected significantly changing oncogenes in pattern 3. Each line represents a patient and colored based on the distance to the median profile.

D   Radial bar plot with percentages of proteins following each pattern (patient-wise pattern) in 35 patients

E   Scatter plots show Spearman rank correlations between percent of proteins following each pattern and relapse-free survival (RFS), based on patient-wise pattern analysis. Significance was determined using t-test, and P values were corrected using Benjamini–Hochberg FDR correction.

Next, to identify the proteins and pathways associated with patient response and relapse, we divided the patients into two groups based on their M&P score (better responders-M&P = 3,4 and worse responders-M&P = 1,2) and performed a paired Student's t-test between matched pre-treatment and post-treatment samples. We identified 316 significantly altered proteins in better responders (q < 0.05); however, 93% of these proteins remained unaltered after treatment in poor responders (Fig 3A). Essentially, significantly downregulated proteins follow pattern 1 in responders and pattern 3 in non-responders, while significantly upregulated proteins follow pattern 2 in responders and pattern 4 in non-responders. Similarly, separate analyses of the relapse groups identified

223 proteins that were differentially expressed in patients with no relapse, while these were unaffected by treatment in patients who showed relapse (Fig 3B). Interestingly, 150 proteins were associated with both drug response and relapse in our cohort. Combined network of upregulated proteins upon treatment in better responders showed a significant enrichment of amino acid biosynthesis pathway, pentose phosphate pathway, inflammatory response, and glycolysis/gluconeogenesis (Fisher exact test, q < 0.05, Fig EV1A). Combined network of downregulated proteins upon treatment in better responders showed a significant enrichment of TCA cycle, oxidative phosphorylation, PPAR signaling, and proline biosynthesis pathway (Fig EV1B).

## Weighted gene correlation network analysis (WGCNA) identifies protein modules associated with neoadjuvant treatment response

To further examine the association of the proteomics data with clinical parameters, we performed unsupervised WGCNA (Storey, 2004; Zhang & Horvath, 2005; Langfelder & Horvath, 2008) on the post-treatment tumor samples, which best reflect tumor response (Appendix Fig S2A, Dataset EV5). Analysis resulted in 39 modules, which were subjected to correlation analysis with clinical features, including age, Ki67 staining, tumor grade, tumor size, relapse, death, survival time, M&P score, and tumor co-clustering (Appendix Fig S2B). Of these, 19 modules correlated with at least one clinical feature (Fig 3C and D; $P < 0.05$, Dataset EV5); six eigen-gene modules correlated with M&P score (Fig EV2A). Three modules, which positively correlated to M&P score, or better responders (pale-turquoise, yellow, and purple modules, collectively referred to as protein cluster A), presented high levels of collagens, integrins, and actin regulators that mediate focal adhesion and cytoskeletal organization (Fig EV2B). In contrast, turquoise, orange, and brown modules (collectively referred to as protein cluster B) negatively correlated with M&P score (poor responders) and was enriched for mRNA processing, components of the ubiquitin-dependent protein catabolic process, spliceosome, fatty acid, and ketone body metabolism ($q < 0.05$; Fig EV3A). Poor responders also showed upregulation of MHC protein complex, and related proteins, including HLA-A, HLA-B, HLA-C, HLA-DR, TAP2, and STAT1 along with interferon signaling post-treatment, suggesting potential involvement of the immune system (see Discussion). Examination of pattern enrichment within these clusters showed that clusters A and B were significantly enriched for pattern 3 and 4 proteins, respectively (global pattern, Fisher's exact test $q < 0.02$). In agreement, clustering of the average protein levels in normal, pre-treatment, and post-treatment samples shows that better responders present dominance of patterns 1 and 2, while poorer responders (MP1 and 2) present dominance of patterns 3 and 4 (Fig EV3B).

To evaluate proteins associated with tumorigenic phenotypes such as tumor size and relapse, we looked at protein networks of cluster C (combined proteins from module blue, white, sienna3, and yellowgreen) that show a positive correlation to these clinical features, and protein cluster D (proteins of module pale-turquoise and red) that shows a negative correlation to tumor size and relapse. Smaller tumors showed increased oxidative phosphorylation and TCA cycle. As expected, large tumors showed higher levels of proliferation markers such as MKI67, EGFR, and MCM complex proteins and elevated glycolysis. This metabolic shift was also accompanied by upregulated pentose phosphate pathway, serine synthesis, and proline biosynthesis (Appendix Fig S3A and B).

## PYCR1 level is associated with drug response and relapse

All bioinformatic analyses described above associate the proline biosynthesis pathway, specifically, PYCR1 and ALDH18A1 with relapse-free survival (global pattern analysis, Fig 2B); relapse (WGCNA, Fig 4A and B, Appendix Fig S3A and supervised analysis, Fig 3B); and treatment response (supervised analysis, Fig 3A, Appendix Fig S4A). PYCR1 and ALDH18A1 are members of the proline cycle that consists of four enzymes that convert glutamate to proline, and two enzymes that catalyze the reverse reactions. Aldehyde dehydrogenase family 18 member A1 (ALDH18A1), also known as pyrroline-5-carboxylate synthetase (P5CS), converts glutamate to $\Delta$1-pyrroline-5-carboxylate (P5C), an intermediate metabolite. P5C is converted to proline by the mitochondrial enzymes pyrroline-5-carboxylate reductase (PYCR1/2) or cytosolic pyrroline-5-carboxylate reductase-like (PYCRL). The $NADP^+$ and $NAD^+$ generated from the biosynthesis reactions are also known to drive glycolysis and the pentose phosphate pathway (Fig 4C; Phang *et al*, 2012). Examination of the levels of all pathway proteins showed that mitochondrial proline biosynthesis pathway proteins, PYCR1, PYCR2 and ALDH18A1, are higher in tumor samples relative to normal tissue before and after treatment. Interestingly, this pattern was very similar across all subtypes in our data (Fig 4D). In contrast, proline degradation enzymes PRODH and ALDH4A1 as well as ornithine aminotransferase (OAT), the bidirectional enzyme that links the proline cycle to urea cycle, were not significantly altered in our data (Appendix Fig S4B).

To validate the proteomic results, we performed immunohistochemical staining of PYCR1 on tissue samples from our cohort. PYCR1 protein level was significantly higher than normal in the cancer samples pre-treatment ($P = 0.002$), and despite some reduction upon treatment, it was still significantly higher than normal also in the post-treatment samples ($P = 0.047$) (Fig 4E and F). Since post-treatment residual cancer is a prognostic factor associated with survival, we used Cox proportional hazard model to determine whether post-treatment PYCR1 level alone had prognostic value in our proteomics data. Survival analysis of our cohort showed that high PYCR1 abundance level (above median) in residual tumors was associated with shorter overall survival (OS) and recurrence-free survival (RFS; hazard ratio OS = 2.4, Cox proportional hazard univariate OS $P = 0.015$; hazard ratio RFS = 2.2, Cox proportional hazard univariate RFS $P = 0.046$; Fig 4G, Appendix Fig S4C). PYCR1 significance remained even after correction for potential confounding factors, including tumor size, grade, and M&P score (Appendix Fig S4D). Interestingly, pre-treatment PYCR1 level was not significantly associated with survival (hazard ratio RFS = 1.2, Cox proportional hazard univariate RFS $P = 0.537$; Appendix Fig S4E). In a multivariate model, proline biosynthesis pathway, containing all four proline biosynthesis proteins, was associated with survival and PYCR1 was the most significant among them (Cox proportional hazard multivariate RFS, PYCR1 $P = 0.037$, global $P = 0.07$; Appendix Fig S4F).

Beyond the clinical significance of PYCR1 in our dataset, we also validated its significance on external datasets. Evaluation of 1,010 treatment-naïve samples from the TCGA breast cancer provisional dataset showed that high mRNA levels of PYCR1 were also significantly associated with poor progression-free survival (Appendix Fig S4G). In addition, a proteomic dataset of drug response in ten patients (Satpathy *et al*, 2020) showed that similar to our observation, PYCR1 was significantly downregulated in samples from complete responders taken 72 h post-treatment and was unaltered in non-responders (Appendix Fig S4H).

## CRISPR/Cas9-based PYCR1 knockout in MDA-MB-231 and MCF7 cells affects tumor growth and drug response

Previously, PYCR1 knockout in triple-negative cell lines was shown to significantly reduce Transwell invasion and tumor burden in mice

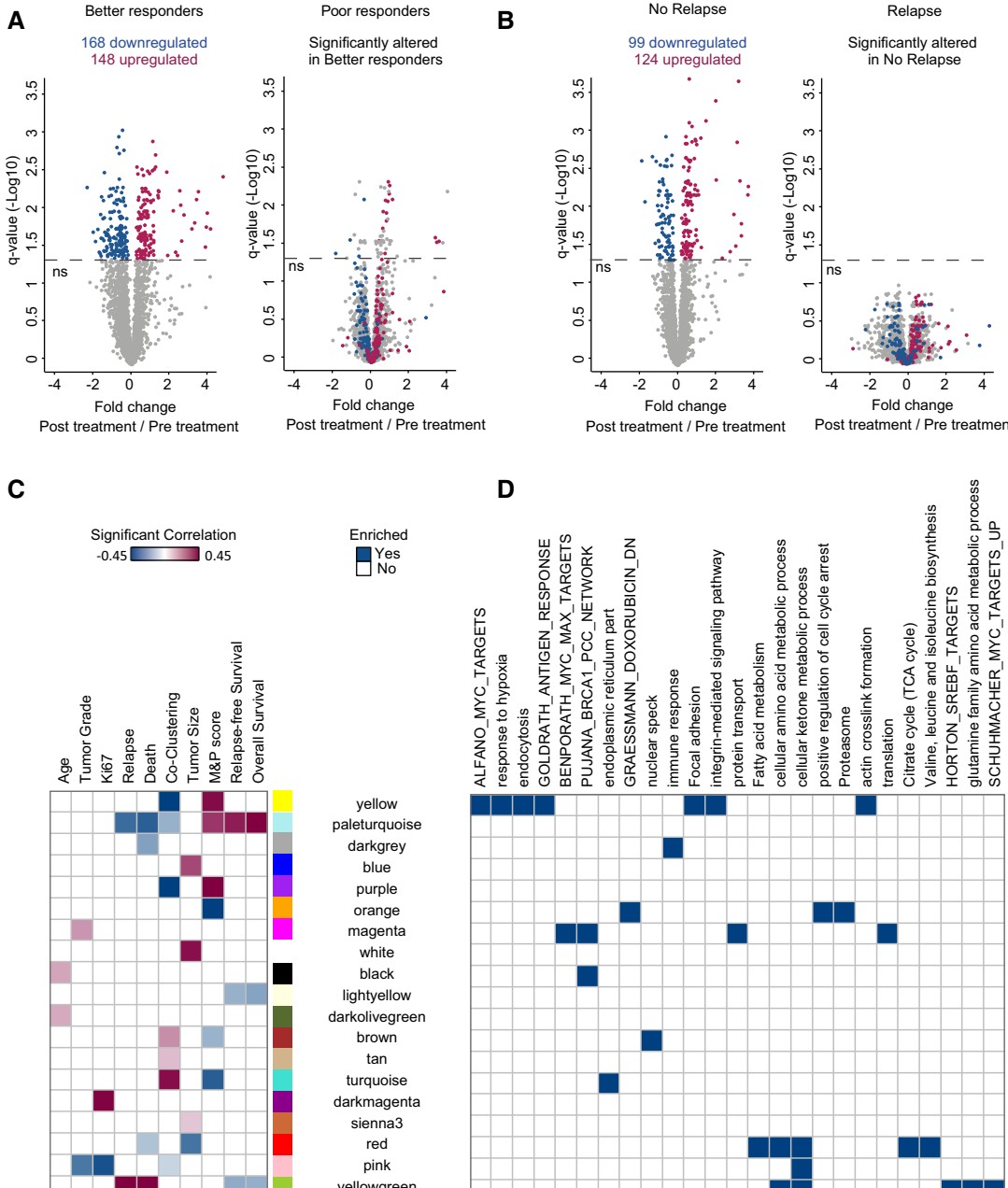

**Figure 3. Supervised analysis and unsupervised WGCNA identifies protein modules associated with neoadjuvant treatment response.**

A  Volcano plot shows significantly altered proteins in better responders before and after treatment. Samples are compared by paired Student's *t*-test, FDR 5%. These proteins are not significantly altered in poor responders.

B  Same as (a) but comparing patients with and without relapse

C  Heat map shows 19 eigengene modules that show significant Pearson correlation (*P* < 0.05) to at least one clinical parameter in post-treatment samples. Age (years); M&P score (1:no reduction in tumor cellularity, 2: minimum reduction in cellularity, 3: partial reduction in cellularity); relapse-free survival time (years); overall survival time (years); tumor grade at diagnosis (1, 2 or 3); relapse (yes/no); death (yes/no); tumor size at diagnosis (1: < 2 cm, 2:2–5 cm, 3: > 5 cm, 4 indicates that tumor has grown into chest wall or skin); Ki67 intensity measured by immunohistochemistry at diagnosis (1–3); co-clustering of matched pre- and post-treatment samples after hierarchical clustering (yes/no, *P* < 0.05).

D  Selected gene ontology processes, KEGG pathways, and gene sets from MSigDB enriched in these modules are indicated (Fisher's exact test, FDR 2%).

(Loayza-Puch *et al*, 2016; Ding *et al*, 2017). We recapitulated these results and showed that knockout of PYCR1 in triple-negative breast cancer cell line MDA-MB-231 reduced invasion and migration

capability, and 2D proliferation *in vitro* (Fig EV4A–D). However, we did not observe significant effects on the response to treatment with paclitaxel and doxorubicin (Fig EV4E). Marked growth inhibition

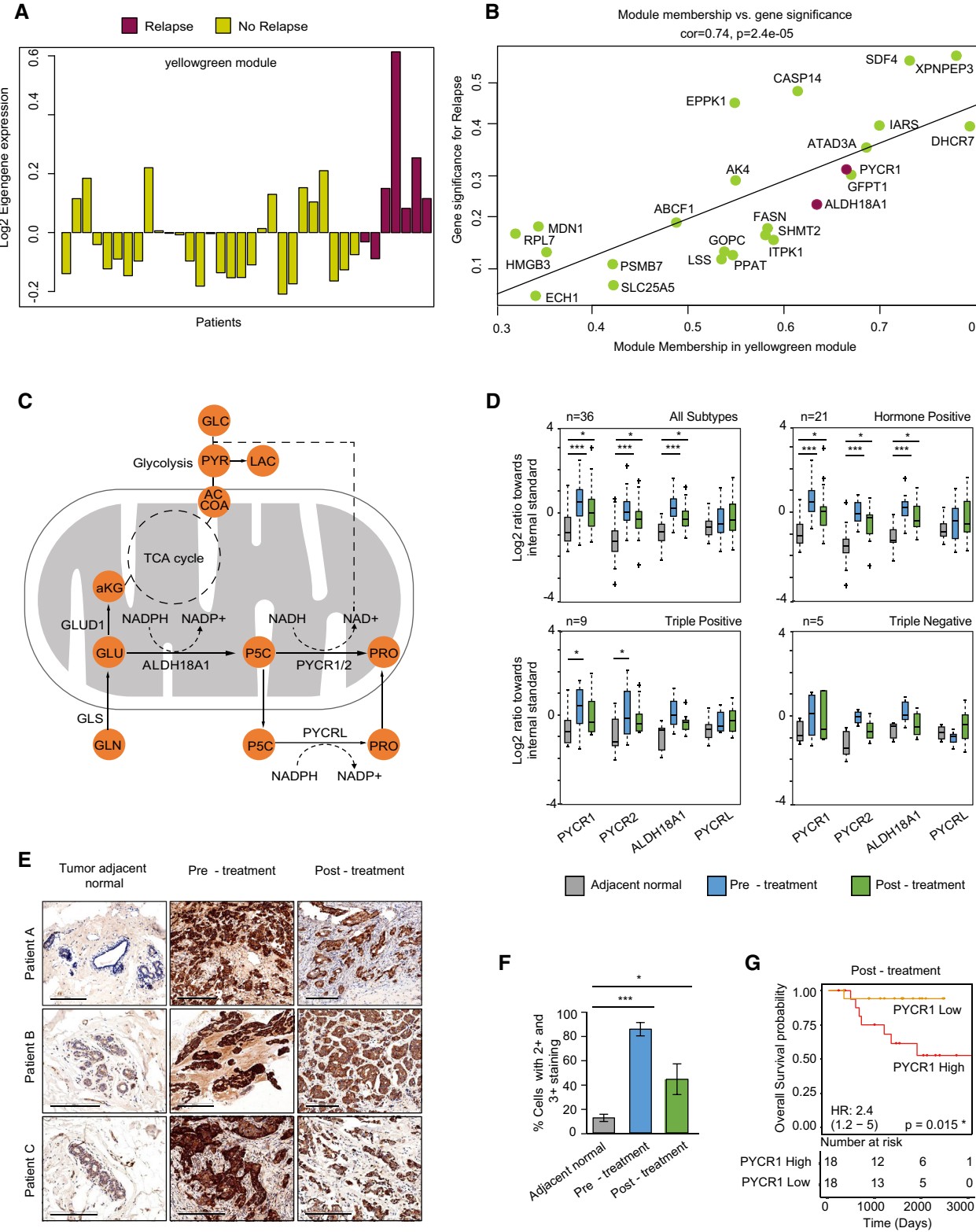

**Figure 4.**

was also observed *in vivo* upon cell injection to the mammary fat pad of immunodeficient mice. We could not evaluate the response to chemotherapy, since tumors did not reach the target tumor size for initiating chemotherapy treatment, even after 26 days (Fig EV4F–H).

Since the pattern of PYCR1 levels was most significant in hormone-positive subtype in our cohort (Fig 4D), we further focused our analyses on the metabolic effects of PYCR1 KO and response to chemotherapy in MCF7 (Appendix Fig S5A). We

**Figure 4. PYCR1 abundance level is associated with breast cancer relapse.**

A  Abundance level of the Eigengene vector of yellowgreen module. Each bar represents a patient and is separated and colored according to relapse status.

B  Scatter plot of module membership vs gene significance score for the yellowgreen module that shows highest correlation to relapse. Proline biosynthesis proteins PYCR1 and ALDH18A1 are highlighted. Significance of Pearson correlation was determined using *t*-test.

C  Diagram of the proline metabolism pathways. Proline biosynthesis genes are PYCRL, pyrroline-5-carboxylate reductase-like; PYCR1, pyrroline-5-carboxylate reductase 1; PYCR2, pyrroline-5-carboxylate reductase 2; ALDH18A1, aldehyde dehydrogenase 18 family member A1 (pyrroline-5-carboxylate synthetase).

D  Proline biosynthesis protein levels in breast cancer subtypes (Log2 ratios of light vs. SILAC standard). The upper and lower limits of the box show values from the first to the third quartile, with the horizontal line indicating the median. The whiskers extend 1.5 times the interquartile range from the edges of the box. Outlier values were plotted outside the whiskers. Significance was determined using paired Student's *t*-test and *P* values were corrected using permutation-based FDR. *$P < 0.05$ and ***$P < 0.001$.

E  High abundance of PYCR1 in the cancer tissue was validated by immunohistochemistry on matched tissue specimens from patients in this study ($n = 3$); scale bar: 200 μm.

F  Quantification of IHC shows percentage of cells with 2+ and 3+ PYCR1 staining intensity. Data are expressed as mean ± SD of three patient samples. Paired samples were compared by Student's *t*-test. *$P < 0.05$ and ***$P < 0.001$.

G  Kaplan–Meier survival curve for PYCR1 levels in residual cancer in the current proteomics data. Cox univariate *P* value, risk table, and hazard ratio with 95% CI indicated.

knocked out PYCR1 by CRISPR/Cas9 and validated the functionality of the KO by extracellular proline measurement (Appendix Fig S5B) and by measurement of incorporation of carbons from $^{13}C_5$ glutamine or $^{13}C_6$ arginine into proline. PYCR1 knockout reduced overall intracellular proline levels and specifically, proline biosynthesis from glutamine (Appendix Fig S5C). Given that the pathway diverts glutamate from the TCA cycle, we examined the effects on central metabolism. Seahorse measurements showed that PYCR1 KO cells present a higher basal respiration rate compared to control cells (Fig 5A); however, the spare respiratory capacity or the ability of cells to maximize mitochondrial respiration during stress was reduced (Fig 5B). This suggests that PYCR1 may play a role in maintaining good mitochondrial function to support maximal respiration under stress, thus presumably contributing to tumor cell survival. In support of these results, flux analysis upon heavy glutamine administration showed that KO of PYCR1 increased incorporation of heavy label into the TCA cycle intermediates fumarate, malate, and citrate in comparison with control cells (Fig 5C). Examination of the glycolytic function showed that extracellular acidification rate (associated with lactate secretion) was significantly reduced upon PYCR1 KO presumably due to low

NAD$^+$ in the cells (Liu *et al*, 2015) (Fig 5D and E). In agreement, measurement of extracellular lactate showed reduced secretion in the KO cells (Fig 5F).

Proteomic analysis of PYCR1 KO and control cells showed significant downregulation of mitochondrial enzyme PYCR2, with no effect on other proline biosynthesis proteins ALDH18A1 and PYCRL (FDR cutoff 0.1; Appendix Fig S5D). Further examination of biological pathways using a non-parametric 1D enrichment test showed significant upregulation of DNA damage response and downregulation of ECM interaction, antioxidant activity, and ABC transporters in the KO cells, suggesting a role in drug response and tumorigenic potential (*q*-value < 0.02; Appendix Fig S5E).

Similar to MDA-MB-231 cells, PYCR1 KO in MCF7 cells significantly compromised their migration and invasion capabilities (Fig 6A and B). Furthermore, PYCR1 KO cells formed smaller and fewer colonies when compared to the control cells under anchorage-independent conditions (Fig 6C). In contrast to MDA-MB-231 cells, we found no effect on the proliferation rate of MCF7 cells in 2D cultures (Fig 6D). In agreement with (Yasuda *et al*, 2013), PYCR1 KO in MCF7 increased sensitivity to oxidative stress, generated by hydrogen-peroxide (Fig 6E).

**Figure 5. CRISPR/Cas9-based PYCR1 knockout increases glutamine flux to the TCA cycle.**

A  Seahorse experiment measuring oxygen consumption rate (OCR) in MCF7 cells upon addition of oligomycin, FCCP, and a mixture of antimycin and rotenone. Arrows indicate the points of addition of each inhibitor. Data are presented as mean ± SD of one representative experiment ($n = 4$ wells). Similar results were obtained in three biological experiments. Samples were compared using Kruskal–Wallis test followed by Dunnett's test for multiple pairwise comparisons. Corrected *P* values were indicated as follows **$P < 0.01$ and ***$P < 0.001$. Shapiro–Wilk test was used to check data normality, and Bartlett test was used to examine homogeneity of variances.

B  Bar plot indicates average basal OCR and spare respiratory capacity in seahorse measurements. Data are presented as mean ± SD of one representative experiment ($n = 4$ wells). Samples were compared using Kruskal–Wallis test followed by Dunnett's test for multiple pairwise comparisons. Corrected *P* values are indicated as follows *$P < 0.05$ and ***$P < 0.001$. Shapiro–Wilk test was used to check data normality, and Bartlett test was used to examine homogeneity of variances.

C  $^{13}C_5$-Glutamine-derived carbon labeling patterns of TCA cycle intermediates fumarate, malate, and citrate in MCF7 cells. Data are presented as mean ± SD of triplicate samples. Isotopologues are represented as M+n where M indicates the mass and n equals number of $^{13}C$ incorporated.

D  Seahorse experiment measuring extracellular acidification rate (ECAR) in MCF7 cells upon addition of glucose, oligomycin, and 2-deoxyglucose (2DG). Arrows indicate the points of addition of each inhibitor. Data are presented as mean ± SD of one representative experiment ($n = 4$ wells). Samples were compared using Kruskal–Wallis test followed by Dunnett's test for multiple pairwise comparisons. Corrected *P* values are indicated as follows **$P < 0.01$. Shapiro–Wilk test was used to check data normality, and Bartlett test was used to examine homogeneity of variances.

E  Bar plot indicates average glycolytic capacity and glycolytic reserve in seahorse measurements. Data are presented as mean ± SD of one representative experiment ($n = 4$ wells). Similar results were obtained in three biological experiments. Samples were compared using Kruskal–Wallis test followed by Dunnett's test for multiple pairwise comparisons. Corrected *P* values are indicated as follows **$P < 0.01$ and ***$P < 0.001$. Shapiro–Wilk test was used to check data normality, and Bartlett test was used to examine homogeneity of variances.

F  Normalized lactate concentration upon PYCR1 KO. Data represents mean ± SE of three biological experiments. Samples are compared using Kruskal–Wallis test followed by Dunnett's test for multiple pairwise comparisons. Corrected *P* values are indicated as follows *$P < 0.05$. Shapiro–Wilk test was used to check data normality, and Bartlett test was used to examine homogeneity of variances.

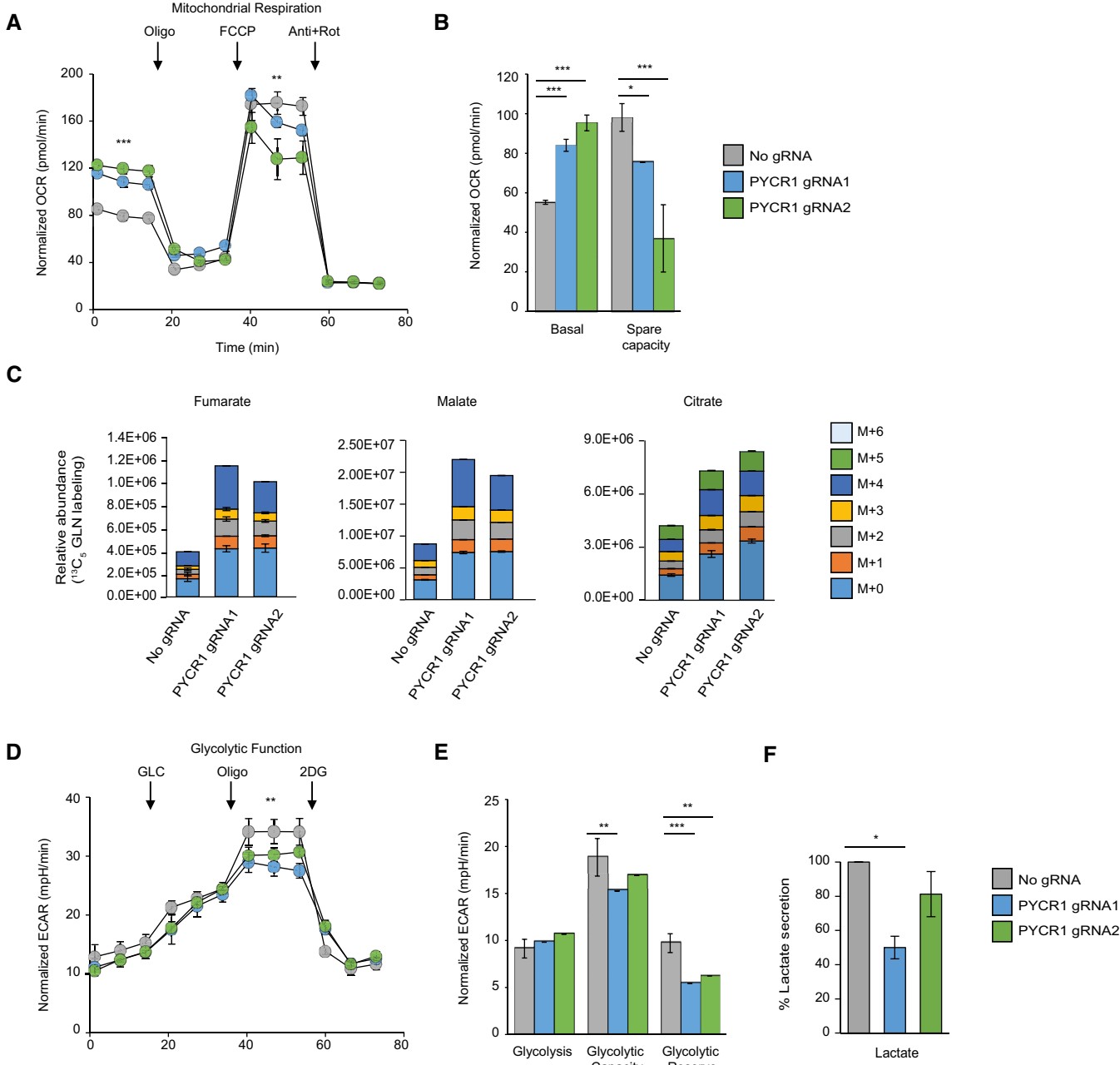

**Figure 5.**

Next, we examined the involvement of PYCR1 in drug response. Measurement of cell survival upon 72 hrs of treatment showed that KO cells were significantly more sensitive to paclitaxel and doxorubicin (Fig 6F and G) and to a lesser extent to cyclophosphamide (Fig 6H), which may result in part, from compromised oxidative stress response in these cells. In agreement with our *in vitro* results, PYCR1-KO tumors induced a slight but significant reduction in tumor size *in vivo* (Fig 6I). Furthermore, WT MCF7 tumors showed no significant difference in tumor weight and volume upon treatment with paclitaxel and doxorubicin (Fig 6J). Finally, in support of our results thus far, PYCR1 KO MCF7 tumors showed a marked reduction in tumor volume and weight upon treatment with two

cytotoxic drugs (Fig 6J, Appendix Fig S6A and B). Collectively, functional perturbations of PYCR1 revealed a role for this metabolic protein in tumor progression and response to treatment and support the observations from our clinical proteomics data, wherein residual tumors with high PYCR1 level are associated with resistance to treatment and with poor clinical outcome.

## Discussion

We present a proteomic study of neoadjuvant treatment response, analyzing a matched cohort of pre- and post-treatment breast cancer

samples and their matched tumor-adjacent normal samples. We took two analytical approaches to associate between protein level dynamics and treatment response. The first, pattern analysis, takes advantage of the matched nature of this cohort and focuses on the change in protein level in the different states. We found that higher

percentage of pattern 3 proteins was associated with shorter relapse-free survival time in patients. Among the pattern 3 proteins, we found multiple oncogenes (e.g., Akt, Stat1, mTOR), and in addition, we found two proline biosynthesis pathway proteins, PYCR1 and ALDH18A1. The second approach focused on the post-treatment

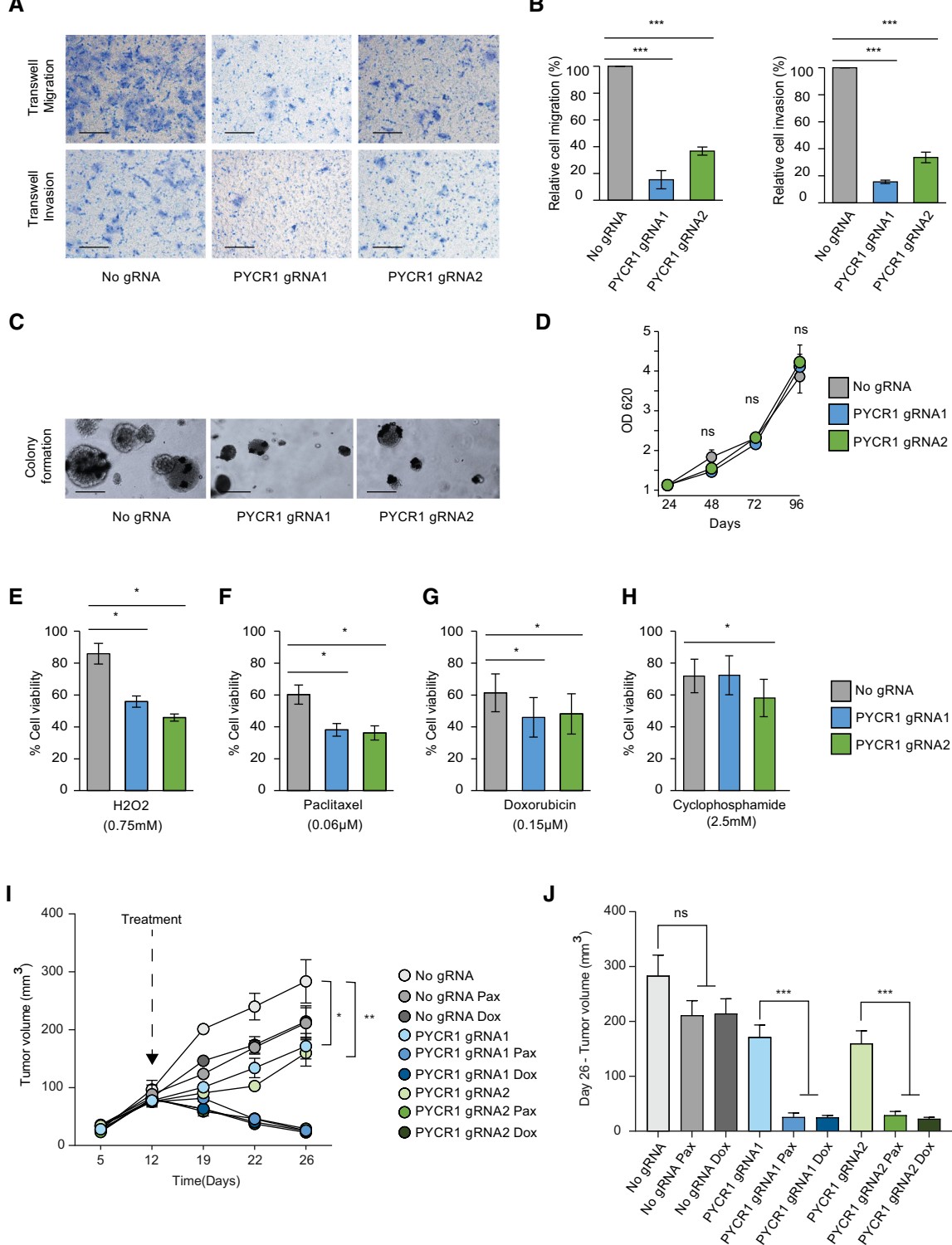

Figure 6.

**Figure 6. PYCR1 knockout results in reduced invasion of MCF7 cells.**

A Representative pictures of Transwell migration and invasion after PYCR1 knockout. Scale bar :100 μm.

B Bar plot represents mean ± SD of three biological replicates for Transwell migration and invasion. Samples were compared using Kruskal–Wallis test followed by Dunnett's test for multiple pairwise comparisons. Corrected $P$ values are reported as follows ***$P < 0.001$.

C Representative images of colony formation in soft agar upon PYCR1 KO. Scale bar: 100 μm.

D Growth measurements of MCF7 wild-type and PYCR1 KO cells over 96 h. Data represent mean ± SE of three biological experiments. Samples were compared using paired Student's $t$-test.

E–H Bar plots show percentage of viable cells after treatment with 0.75 mM $H_2O_2$ (E), 0.06 μM paclitaxel (F), 0.15 μM doxorubicin (G), and 2.5 mM cyclophosphamide (H). Data represent mean ± SE of three biological experiments. Samples are compared using Kruskal–Wallis test followed by Dunnett's test for multiple pairwise comparisons. Corrected $P$ values are indicated as follows *$P < 0.05$.

I Tumor volume measurements for 26 days in MCF7-injected NSG mice. CRISPR control and PYCR1 KO tumors with ($n = 8$) and without treatment ($n = 5$) are shown. Data represent mean ± SE. Pax: paclitaxel, Dox: doxorubicin. Groups were compared by one-way ANOVA followed by Tukey's multiple comparisons test for pairwise group comparisons. Corrected $P$ values are reported as follows *$P < 0.05$, **$P < 0.01$. Shapiro–Wilk test was used to check data normality, and Bartlett test was used to examine homogeneity of variances.

J Bar plot indicates mean ± SE of tumor volume measurements for day 26. CRISPR control and PYCR1 KO tumors with ($n = 8$) and without treatment ($n = 5$) were compared by one-way ANOVA followed by Tukey's multiple comparisons test for pairwise group comparisons. Corrected $P$ values are reported as follows ***$P < 0.001$. Shapiro–Wilk test was used to check data normality, and Bartlett test was used to examine homogeneity of variances.

samples, which are associated with response and relapse. These results largely overlapped with the first approach and reinforced the importance of PYCR1 and ALDH18A1. Kaplan–Meier analysis using Cox proportional hazard model showed that indeed patients with high PYCR1 level in post-treatment residual carcinoma had poorer prognosis than patients with low PYCR1 level.

The clinical proteomic results identified PYCR1 as a candidate cancer regulator irrespective of cancer subtype, while having stronger association with hormone-positive tumors. In agreement, we found that PYCR1-KO affects the invasive migratory capacity in both MCF7 cells (ER$^+$) and MDA-MB-231 cells (TNBC). These results are further reinforced by the proteomic analyses of PYCR1-KO cells, which suggest reduced ECM receptor interactions with downregulation of integrins and laminins. In contrast to the common PYCR1 effects, we found that in MDA-MB-231 cells, the KO affects growth *in vitro* and *in vivo*, while in MCF7 cells, growth rates were hardly affected. However, while the PYCR1-KO had no effect on chemotherapy-treated TNBC cells, we observed marked increase in response to chemotherapy in ER$^+$ cells and tumors. Considering the relatively low sensitivity of ER$^+$ tumors to chemotherapy, finding their vulnerability to this treatment holds a promising clinical implication. Given the marked effect of the KO on central metabolism, we associate the different effects on the basal metabolic differences between TNBC and ER$^+$ tumors (Tyanova *et al*, 2016a; Yanovich *et al*, 2018). While TNBC presents high glycolytic activity, ER+ tumors present higher oxidative metabolism. Therefore, the effect of PYCR1-KO, which also reduces glycolytic flux, is more specific to TNBC cell growth.

Apart from the metabolic effects, mutations in PYCR1 and ALDH18A1 have been shown to affect elastin and collagen formation in the extracellular matrix, leading to progeroid changes and wrinkled skin, known as Cutis Laxa syndrome (Scherrer *et al*, 2013). Interestingly, extracellular matrix stiffness was shown to correlate with pathological response to NAT (Evans *et al*, 2013). Altogether, our results add another layer to the understating of proline biosynthesis in the context of tumor aggressiveness and relapse, and place this pathway as a central mediator of metabolism, cancer cell invasion, and drug response.

Our results open a new gateway for potential combination of chemotherapy with PYCR1 inhibitors, for the treatment of ER$^+$ and TNBC, as well as heterogeneous tumors that contain TN regions adjacent to ER$^+$ regions. This could also be extended to other cancers where PYCR1 is known to be upregulated. In support of this hypothesis, meta-analysis of 13 pooled microarray datasets showed better survival of stage II breast cancer patients with low PYCR1 expression (Ding *et al*, 2017). Furthermore, upregulation of PYCR1 has been shown to be associated with increased tumorigenic and metastatic potential in several cancer types (De Ingeniis *et al*, 2012; Elia *et al*, 2017; Zeng *et al*, 2017; Ye *et al*, 2018; Wang *et al*, 2019).

Beyond PYCR1, our analysis uncovers many proteins with a potential role in mediating drug resistance and can uncover new therapeutic opportunities. For example, investigation of cluster A proteins, which were associated with poor chemotherapy response, showed high levels of MHC processing, presentation proteins, and lipid metabolism proteins (Appendix Fig S3B). Recently, in a cohort of melanoma response to anti-PD1 immunotherapy, we showed that tumors of responders expressed higher mitochondrial lipid metabolism proteins, which led to elevated antigen presentation and interferon signaling (Harel *et al*, 2019). High levels of these proteins in poor responders to chemotherapy suggest that these may be responsive to immunotherapy. Despite disappointing results of breast cancer response to single-agent immunotherapy, recent clinical trials suggest higher response of triple-negative tumors in combination with chemotherapy (Kim *et al*, 2019; Planes-Laine *et al*, 2019). Our results suggest that these approaches may also be applicable to aggressive ER$^+$ tumors that do not respond to neoadjuvant therapy. Altogether, this proteomic resource may be the basis for multiple functional and translational studies aiming to identify the cancer vulnerabilities and increase therapeutic responses.

# Materials and Methods

### Cohort assembly

113 formalin-fixed paraffin-embedded (FFPE) samples were retrospectively collected from 35 women with breast cancer who showed partial response to NAT and from five healthy women who underwent breast reduction surgeries. The study was approved by the Institutional Review Board of Sheba Medical Center (Approval No. 8736-11-SMC) and Tel Aviv University, with full exemption for

consent form for anonymized samples. All samples were anonymized as defined in the study protocol. All patients in the cohort had stage II or III invasive breast cancer. 33 patients had grade 2 and 3 tumors, and 29 patients showed lymph node involvement. All patients underwent NAT to reduce tumor burden with AC-T (four cycles of doxorubicin (60 mg/m$^2$) and cyclophosphamide (600 mg/m$^2$) every 2 weeks followed by paclitaxel (175/m$^2$) two weekly for four cycles. Her2 overexpressing patients also received Herceptin. For every patient in the cohort, we compared the reduction in tumor cellularity before and after treatment and calculated the Miller & Payne (M&P) pathological response score (MP1: no reduction in tumor cellularity/poor responders, $n = 6$; MP2: minimum reduction in cellularity/partial responders, $n = 13$; MP3: partial reduction in cellularity/better responders, $n = 16$; MP4: significant reduction in cellularity/better responders, $n = 1$). The single MP4 patient was analyzed together with the other better responders as MP3 ($n = 17$).

## Cell lines

For *in vitro* functional analysis and proteomics, MCF7 and MDA-MB-231 cell lines were cultured in DMEM-high glucose media without proline (Biological industries). Media were supplemented with 10% dialyzed FBS (Biological industries), 1 mM sodium pyruvate, and 1mM penicillin–streptomycin at 5% $CO_2$. MCF7 cell line was authenticated by analyzing STR profiles using Promega PowerPlex 16 HS kit at the Genomics Core Facility of BioRap Technologies and the Rappaport Research Institute in Technion, Israel. Cell cultures were routinely verified to be mycoplasma-free by PCR-detection kit (Hy-Mycoplasma Detection Kit, Biolabs). For exo-metabolome analysis, the MCF7 cells were cultured in glucose/glutamine/proline-free DMEM (Biological Industries), supplemented with 10% DFBS, 1% antibiotics, 2 mM glutamine, and 25 mM glucose.

## MS-based proteomics

Formalin-fixed paraffin-embedded tissue samples were deparaffinized with xylene according to the standard protocol and rehydrated in graded ethanol. Using hematoxylin and eosin stained tissue samples as template, tumor and tumor-adjacent normal ducts were macrodissected to contain at least 70% cellular areas. Pathological analysis of each tumor ensured that we avoid mainly fibrotic and necrotic areas, as well as avoid regions with intense lymphocyte infiltration and adipose tissue. Analysis of core needle biopsies provides a very unique sample type, but also limited in protein amounts. For each biopsy sample, we macrodissected cancer cell regions from 2 to 3 FFPE sections (~5- to 10-mm$^2$ tissue area in each section) of 10 μm thickness, resulting in ~7.5 μg protein amounts. Post-treatment samples were extracted from 100-mm$^2$ tissue area. The scraped tissues were lysed in 50% trifluoroethanol (TFE) in 50 mM ammonium bicarbonate buffer followed by heating for 1 h at 95°C and sonication for 10 cycles (30 s on, 30 s off) in a Bioruptor Sonicator (Diagenode). After 20 min high-speed centrifugation, the supernatant was transferred to a new tube and protein amounts were calculated by Coomassie Brilliant Blue assay (Minamide & Bamburg, 1990). 7.5 μg total protein in each sample was mixed 1:1 with the breast super-SILAC mix that served as an internal standard (Geiger et al, 2010). Proteins were reduced with 5 mM dithiothreitol (DTT) and alkylated with 15 mM IAA (Iodoacetamide) followed by

overnight in-solution digestion at 37°C with Lysine-C (LysC)-Trypsin mix (Promega, 1:100 enzyme:protein ratio) and sequencing grade-modified trypsin (Promega, 1:50 enzyme: protein ratio), respectively. The resulting peptides were acidified with trifluoroacetic acid (TFA) and separated using strong cation exchange (SCX) fractionation (Wiśniewski et al, 2009). Peptides were desalted on $C_{18}$ stage tips (Rappsilber et al, 2003), vacuum dried, and resuspended in 2% acetonitrile/0.1% TFA.

For cell line proteomic experiments, MCF7 cells were cultured in DMEM (without proline) for 72 h and lysed with 6 M urea/2 M thiourea in 50 mM ammonium bicarbonate buffer in three biological repeats (3 technical repeats each). Protein concentrations were measured using Bradford assay, and in-solution digestion was performed with LysC-Trypsin mix (1:100 enzyme: protein ratio) and trypsin (Promega; 1:50 enzyme: protein ratio). Peptides were desalted on $C_{18}$ stage tips, vacuum dried, and resuspended in 2% acetonitrile/0.1% TFA.

Samples were analyzed by liquid-chromatography using the EASY-nLC1000 HPLC (Thermo Fisher Scientific) coupled to either Q-Exactive (QE) Plus or Q-Exactive HF mass spectrometers (Thermo Fisher Scientific, Bremen, Germany). Peptides were separated on 75 μm × 50 cm long EASY-spray PepMap columns (Thermo Fisher Scientific) and loaded with Buffer A (0.1% formic acid). Peptides were eluted with a gradient of 5–28% Buffer B (80% acetonitrile/0.1% formic acid), at a flow rate of 300 nl/min, over a gradient of 140 min. MS acquisition was performed in a data-dependent manner, with selection of the top 10 (for QE-Plus) and top 15 (for QE-HF) most intense peaks from each MS scan for fragmentation at MS/MS level.

## MS-based metabolomics

For endo-metabolome analysis, MCF7 control and PYCR1-KO cells were cultured in DMEM (without proline) until 80% confluence. Medium was aspirated, and cells were gently washed once with PBS. Medium was replaced with glucose or glutamine-free DMEM (Biological Industries), supplemented either with 2 mM $^{13}$C-glutamine or 1 mM $^{13}$C-arginine (Cambridge Isotope Laboratories), 10% DFBS, 1% antibiotics for 4 h. Endo-metabolome was extracted with methanol:acetonitrile:water (5:3:2) on top of a dry ice-ethanol bath, and the lysates were rotated at 4°C for 10 min and centrifuged at 14,000 $g$ for 10 min at 4°C. Supernatants were stored at −80°C. Experiments were conducted in three biological replicates and three technical replicates each.

For exo-metabolome analysis, MCF7 control and KO cells were cultured for 72 h in glucose/glutamine/proline-free DMEM (Biological Industries), supplemented with 10% DFBS, 1% antibiotics, 2 mM glutamine, and 25 mM glucose. Medium without cells served as a control. Exo-metabolome was extracted by transferring 50 μl of medium into 750 μl ice-cold extraction solution of acetonitrile: water (4:1), vortexed vigorously, and centrifuged at 14,000 $g$ for 10 min at 4°C. Supernatants were stored at −80°C. Three biological replicates were performed, with three technical replicates each. Protein concentration was calculated using Bradford assay for data normalization.

Metabolites were separated on SeQuant ZIC-pHILIC (150 × 2.1 mm, 5 μm) column coupled to a SeQuant ZIC-pHILIC guard column (20 × 2.1 mm, 5 μm) (Merck) with flow rate 0.1 ml/min. LC-MS analysis was performed on an Ultimate 3000 UHPLC

(Thermo Scientific). QE-Plus mass spectrometer (Thermo Scientific) operated in a polarity switching mode (between positive and negative ion modes). Metabolites were separated in a 49-min gradient of acetonitrile and 50 mM ammonium carbonate (pH10). Injection volume was 5 μl.

## CRISPR/Cas9 knockout of PYCR1

Two single guide-RNAs TGAAATAGGCGCCGACATTG (Loayza-Puch *et al*, 2016) and TCTCCGGACAGCATGAGCG (Sanjana *et al*, 2014) were cloned into PX459 vector (Addgene plasmid # 62988; Ran *et al*, 2013), which contains the Cas9 enzyme from S. pyogenes. The cloned vectors were transfected into MCF7 cells with Xfect transfection reagent (Takara bio) and treated with puromycin for 2–3 weeks followed by Western blot to confirm KO. CTTCATCGGCGCTGGC CAGC gRNA was used to create the KO MDA-MB-231 cell line. Empty vector without the gRNA served as control.

## Western blot

Whole cell lysates (20 μg) from MCF7 and MDA-MB-231 were separated by precast SDS-PAGE (Bio-Rad). Western blots were reacted with the following antibodies: Rabbit anti-PYCR1(1:1,000; Protein-Tech, #13108-1-AP) and Mouse anti-Tubulin (1:400,000; Sigma, #T6074) at 4°C, overnight. Secondary antibodies were conjugated to horseradish peroxidase.

## Cell proliferation and viability assays

Control and KO MCF7 cells were seeded in triplicates at a density of 2,500 cells/well in 96-well plates for cell proliferation measurement. Cells were fixed with 0.5% glutaraldehyde every 24 h for 4 days, and growth rates were assessed by methylene blue assay. For drug sensitivity assay, 5,000 cells/well were seeded in a 96-well plate in triplicates and treated with the drugs at a concentration that showed 50% cell viability. Briefly, after 24 h, the cells were treated with either 0.06 μM paclitaxel, 0.15 μM doxorubicin, or 2.5 mM cyclophosphamide. Cells were fixed after 72 h and assessed by methylene blue assay. For the oxidative stress rescue test, cells were treated with 0.75 mM $H_2O_2$ for 48 h. Experiments were performed in biological triplicates. Similarly, control and KO MDA-MB-231 cells were seeded in triplicates at a density of 2,000 cells/well in 96-well plates for cell proliferation measurement and the cells were fixed everyday for 4 days. For drug sensitivity assay, 10,000 cells/well were seeded in a 96-well plate in triplicates and treated after 24 h with either 1.25 μM paclitaxel or 0.314 μM doxorubicin.

## Soft agar assay

Anchorage-independent growth was examined by growing MCF7 cells embedded in agarose and on top of a dense agarose layer. Base agar (1 ml of 1% agar) was added to each well of a 6-well plate and allowed to solidify at room temperature. Then, 15,000 cells/well were prepared in 0.3% agarose and layered on top of the base agar. 1ml of medium was added on top of the two agar layers. Cells were grown for 2 weeks in a humidified incubator at 37°C. Colonies were fixed with 4% paraformaldehyde, stained with 0.005% crystal violet in ethanol for 30 min, and photographed. Three independent experiments were performed.

## Transwell assays

Migration and invasion assays were performed by plating $1 \times 10^5$ MCF7 cells and $8 \times 10^4$ MDA-MB-231 cells in Transwell inserts (8 mm pore size, BD Biosciences), on top of Matrigel (ECL cell attachment matrix, Millipore) for invasion assay, or without Matrigel for migration assay. Cells were cultured in serum-free DMEM, and complete medium with 10% serum was added to the bottom chambers. Cells were fixed with 2.5% glutaraldehyde and stained with 1% methylene blue. MCF7 cells were fixed after 48 h, and MDA-MB-231 cells were fixed after 12 h. Top (non-migrating) cells were gently removed, and migrating cells were photographed. Assays were performed in biological triplicates.

## Immunohistochemistry

Matched FFPE specimens from three non-responders to NAT were obtained from the Institute of Pathology, Sheba Medical center. Tissue sections (3.5 μm) were probed with anti-PYCR1 antibody (1:100, ProteinTech #13108-1-AP) using BOND-RX automated staining platform (Leica Biosystems) following selected protocol for the Bond Polymer Refine Detection Novocastra kit (catalog number DS9800, Leica Biosystems). Slides were scanned using Aperio ScanScopeXT system (Leica Biosystems) with selected objective of 20×. Tissue areas with normal or tumor regions were manually annotated and subsequently analyzed by optimized cytoplasm algorithm (Leica Biosystems). Percentage of positively stained cells (intensity 2+ and 3+) were used for statistical analysis.

## Seahorse metabolic analysis

Seahorse-XF 96-well plates were coated with 50 μg/ml Poly-D-lysine for an hour. MCF7 control and KO cells were seeded at a density of 15,000 cells/well and incubated for 24 h at 37°C in 5% $CO_2$ atmosphere. The Mito Stress Test Kit and Glyco Stress Test Kit (Agilent Technologies, Santa Clara, CA, USA) were used to measure oxygen consumption rate (OCR) and extracellular acidification rate (ECAR), respectively. For OCR measurement, the culture medium was replaced with 180 μl of bicarbonate-free and phenol red-free DMEM (Seahorse base medium #103335) incubated without $CO_2$ for 1 h before the assay. The base medium was supplemented with 2 mM L-glutamine, 10 mM glucose, and 1 mM sodium pyruvate; pH was adjusted to 7.4. For serial injections, we used oligomycin (1 μM), carbonyl cyanide-4-(trifluoromethoxy) phenylhydrazone FCCP (1 μM), antimycin A (0.5 μM), and rotenone (0.5 μM). For ECAR measurement, the base assay medium was supplemented with 2 mM L-glutamine and pH was adjusted to 7.4. Cells were pre-incubated for 1 h without $CO_2$ before starting the assay. Sequential injections were performed with glucose (10 mM), oligomycin A (1 μM), and 2-deoxy-D-glucose (2- DG; 50 mM). Both, oxygen consumption and extracellular acidification, were measured using the Seahorse XF96 Extracellular Flux Analyzer (Agilent, CA, USA) for 90 min as described (Invernizzi *et al*, 2012). ECAR and OCR measurements were normalized to cell number, obtained by methylene blue assay.

## Animal experiments

All animal procedures were approved by the Institutional Animal Care and Use Committee of the Weizmann Institute of Science. Female NSG mice (6 weeks old) were obtained from the in-house Weizmann Institute colony. 17 beta-estradiol (E2) pellets (0.72 mg/pellet, 30-day release time; Belma Technologies, Belgium) were implanted underneath the back skin. Two days later, $1 \times 10^7$ MCF7-WT (no gRNA) or MCF7-PYCR1 KO cells (gRNA1 and gRNA2) were inoculated into the mammary fat pad under anesthetic conditions. Once the palpable tumor was observed, mice were randomly divided into treatment and no-treatment group. Animals were then treated with paclitaxel (16 mg/kg) or doxorubicin (8 mg/kg) intravenously (IV) twice a week. Tumor width (W) and length (L) were measured once a week using a caliper, and tumor volume (V) was calculated according to the following formula: V = (W/2× L/2) × 3.14 × 1.33. Body weight was evaluated once per week. At the end of the study, mice were euthanized, tumors were photographed, and weight was recorded. Similarly, $1 \times 10^6$ MDA-MB-231 cells (WT-no gRNA and gRNA1) were implanted to mammary fat pad of NSG mice and tumor growth was monitored without treatment. Treatment groups of MCF7-injected mice were compared with one-way ANOVA, followed by Tukey's multiple comparisons test for pairwise group comparisons. Corrected P values are reported. MDA-MB-231 WT and PYCR1 KO tumors were compared with paired Student's t-test, and P value is indicated.

## Quantification and statistical analysis

### MS-based Proteomics and data filtration

Raw files were analyzed using MaxQuant software (version 1.5.5.1) with integrated Andromeda search engine (Cox & Mann, 2008; Cox et al, 2011). MS/MS spectra were searched against the human FASTA file from the UniProt database (September 2015), a reverse decoy database and common contaminants (a list of 245 entries). The peptide search included cysteine carbamidomethylation as a fixed modification, and N-terminal acetylation and methionine oxidation as variable modifications. Trypsin was selected as the specified protease, and maximum of two missed cleavages were allowed. The minimal peptide length was set to seven amino acids, and "match between runs" feature was enabled. A false discovery rate cutoff of 1% was applied at both the protein and PSM identification levels.

The Perseus program (versions 1.5.5.3 and 1.6.2.1; Tyanova et al, 2016b), MATLAB (version R2016a) and R (http://cran.r-project.org/) were used for statistical analysis. The protein groups were filtered to remove potential contaminants, peptides matched to the reverse decoy database, and proteins only identified by a modification site resulting in 7,600 protein identifications and 7,180 quantifications. Next, the H/L normalized data were $\log_2$ transformed and converted to L/H ratios. These data were filtered to have at least 2/3 quantified values per matched sample (per patient), and if the $3^{rd}$ value was missing, it was imputed separately based on normal distribution with a width of 0.3 and a downshift of 1.2 standard deviations. Patient-wise filtration and imputation ensured that we do not compare matched protein levels of a patient based on 2/3 missing values. On average, every sample had 4.5% of imputed values at the end.

The entire list of 7,600 identified proteins (after patient-wise filtration and imputation) was used for patient-wise pattern analysis and to calculate the percentage of patient-wise patterns in each patient individually. A fold change threshold of 1.5 was used between matched samples to define the different patterns and reflects the trend of changes between normal, pre-, and post-treatment samples.

This entire dataset was further filtered to have quantified values (valid values) in at least 70% of samples. Principal component analysis was used to normalize the data for batch effects by subtracting the $3^{rd}$ component. The resulting dataset of 2,915 proteins was used to perform global pattern analysis with paired Student's t-test to integrate all samples in the cohort and was used to show general functional associations with response. The global pattern analysis was conducted by performing three paired Student's t-tests between each of the three groups for each protein with 5% FDR and S0 = 0.3.

### Co-clustering analysis

Data clustering was performed in R using Spearman correlation as distance measure and average linkage. P value for each cluster in the co-clustering analysis was generated with the R package "pvclust" via multiscale bootstrap resampling of 10,000 iterations.

### Centrality analysis

Protein interaction network of 736 pattern 3 proteins was imported from STRING database, and connected nodes were visualized in Cytoscape. Betweenness centrality of nodes was calculated using the "cytoNCA" tool in Cytoscape. Node size and color of top 10% of the network were adjusted based on centrality value.

### Weighted gene correlation network analysis

Weighted gene correlation network analysis was performed with the R package to extract subnetworks associated with pathological response score and relapse. This method identifies clusters of highly correlated genes/proteins across the samples, and calculates a module eigengene vector for each cluster. The module eigengene enables the user to relate modules to one another and to external sample traits. The R package also provides the gene significance (GS, correlation between each protein in the module to the clinical trait) values and module membership (MM, the correlation of each protein in the module to the module eigengene) values of genes/proteins in each module. We used an input matrix consisting of 2,915 proteins from post-treatment samples. Soft thresholding power was set to 10, and "signed" network function was selected. Pearson correlation was calculated between module eigengene and the following clinical parameters: Ki67 intensity, tumor grade at diagnosis, age, relapse, death, relapse-free survival time, overall survival time, tumor size at diagnosis, and Miller & Payne score. In addition, we associated the eigengene profiles with co-clustering of matched pre- and post-treatment samples after hierarchical clustering. P values are indicated. Network of proteins combined from different modules was constructed using the STRING database (http://string-db.org). All networks were visualized with Cytoscape (http://www.cytoscape.org/). Node size was set based on degree of connectivity of each protein to other interacting proteins (minimum 1, maximum 88).

### Enrichment analysis

For supervised and unsupervised WGCNA analyses, gene annotations including GOBP, GOMF, GOCC, GSEA gene sets from MSigDB, and KEGG pathway were added from UniProt, and Fisher exact test was performed with an FDR cutoff of 2% or 5% (as indicated for each analysis) with the entire identified data of 7,600 proteins as background.

### Survival analysis

Univariate Cox proportional hazard and Kaplan–Meier analysis were performed with R package "survminer". For survival analysis in our dataset, all 35 patients were included and patients were stratified based on median protein level of PYCR1 (above median = high, below median = low). Multivariate Cox proportional hazard analysis was performed with R package "survival". Both global and univariate $P$ values are indicated. Number of events in the multivariate model was 7, and all 35 patients are included. For analysis using mRNA expression, we used the cBioportal database to mine TCGA invasive breast cancer provisional dataset of 1,010 samples. High or low mRNA expression was determined by the number of standard deviations (SD) from the mean (mRNA expression z-Scores of RNA Seq V2 RSEM values).

### Cell line proteomics

For analysis of PYCR1 WT versus KO MCF7 cell line, we used the label-free algorithm in MaxQuant (version 1.5.6.9) for relative quantification. $Log_2$ data were filtered to retain only proteins with 70% valid values across samples. Missing data were overcome by imputing values based on normal distribution with a width of 0.3 and a downshift of 1.2 standard deviations. Differentially expressed proteins between WT and PYCR1 KO MCF7 cells were extracted by performing Student's $t$-test (Benjamini–Hochberg FDR 0.1) after batch-effect removal using "Limma" package in R. 1D annotation enrichment test was performed on the $t$-test difference (Benjamini–Hochberg FDR, $q$-value $< 0.02$).

### MS-based metabolomics

Metabolites were identified and analyzed with Xcalibur and LCquan 2.7 (Thermo scientific) based on retention time and mass-to-charge ratio with 20 ppm threshold. Retention time was calculated based on a calibration curve of metabolite standard. Peak areas of the metabolites were normalized to the intensity of the MS raw files and to the total protein amount. Exo-metabolites were also normalized to the media control (without cells). For $^{13}C_6$-glutamine labeling experiments, heavy carbon incorporation results in isotopologues of metabolites. The increase in mass (M) is reported as M+0 (all carbons unlabeled, *e.g.*, $^{12}C$) to M+6 (all carbons labeled, *e.g.*, $^{13}C_6$), respectively.

## Data availability

All raw files and clinical proteomics data can be downloaded from ProteomeXchange Consortium via the PRIDE repository (project number: PXD012000; http://www.ebi.ac.uk/pride/archive/projects/PXD012000).

**Expanded View** for this article is available online.

## Acknowledgements

We thank the members of the Geiger laboratory for fruitful discussions and technical assistance. We thank Mariya Mardamshina for help with Immuno-histochemistry, Dr. Gilgi Friedlander for bioinformatic advice, and Dr. Einav Nili Gal-Yam for clinical advice. We thank Ofer Barnea Yizhar for providing px459 plasmid and advice on CRISPR/Cas9 system. We thank Constantiner Institute for the partial support. The project was supported by funding from the Israel Ministry of Science and Technology grant 3-11175 and funding from the Horizon2020 ERC grant 639534.

## Author contributions

AS performed all the proteomic analyses, analyzed the clinical samples, and performed the functional *in vitro* experiments; BK, MD, NBo, and TG designed the study; NBN performed the *in vivo* experiments; FLP and RN and RA generated the PYCR1 KO MDA-MB231 cell line; GP assembled the clinical cohort; IM, NBa, and IB performed the pathological analysis of the samples; YY supervised the *in vivo* experiments; AP provided the clinical sample. TG supervised the study; TG and AS wrote the manuscript.

## Conflict of interest

The authors declare that they have no conflict of interest.

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
