## [Review Process File · Molecular Systems Biology]

Proteomic patterns associated with response to breast cancer neoadjuvant treatment

Anjana Shenoy, Nishanth Belugali Nataraj, Gili Perry, Fabricio Loayza Puch, Remco Nagel, Irina Marin, Nora Balint, Noa Bossel, Anya Pavlovsky, Iris Barshack, Bella Kaufman, Reuven Agami, Yosef Yarden, Maya Dadiani, Tamar Geiger

DOI: [10.15252/msb.20209443](https://doi.org/10.15252/msb.20209443)

Corresponding author: Tamar Geiger (geiger@tauex.tau.ac.il)

Review Timeline:

Submission Date:	8th Jan 20
Editorial Decision:	6th Feb 20
Revision Received:	23rd Jul 20
Editorial Decision:	14th Aug 20
Revision Received:	24th Aug 20
Accepted:	31st Aug 20

Editor: Maria Polychronidou

Transaction Report:

Thank you again for submitting your work to Molecular Systems Biology. We have now heard back from the three referees who agreed to evaluate your study. Overall, the reviewers think that the presented findings are potentially interesting. They raise however a series of concerns, which we would ask you to address in a revision.

Without repeating all the points listed below, some of the more fundamental issues raised are the following:

- Both reviewers #1 and #3 think that in vivo validations (using a mouse model) of the role of PYCR1 would need to be included.
- A more thorough statistical analysis should be performed to better support the main conclusions.
- Reviewer #2 mentions that the discussion needs to be streamlined. Moreover, the clinical implications of the presented findings need to be highlighted more clearly in the Discussion.

Reviewer #1:

The manuscript "Proteomic patterns associated with response to breast cancer neoadjuvant treatment" is interesting and presented with good language and a nice flow. The proteomic data and the comparison of the three tissue types (adjacent normal, pre-treatment, post-treatment) are well done. However, Shenoy et al. should be more thorough with the statistical tests they use. Often, p-value adjustment was not done after multiple testing, especially for the WGCNA analyses. Multiple groups should not be compared with multiple t-tests. Also, the statistics should be presented more thoroughly in the methods section, especially the survival analyses. In addition, the most important finding of the authors, namely the important role of PYR1, is not validated by in vivo experiments in breast cancer mouse models to support their findings and supplement their proteomics bioinformatics hypothesis.

Major

- The possible role of PYCR1 loss of function for breast cancer tumor growth should be validated in

an in vivo setting, especially in combination with the chemotherapeutics used in fig. 6e-i

- Figure 4b; WGCNA: What are the adjusted p-values for gene significance (GS) and module membership (MM) in the yellowgreen module for PYCR1 and ALDH18A1, respectively? Was an adjusted-p-value cutoff used to select only proteins with a significant GS and MM?
- WGCNA : On what basis were the modules merged? It is not stated in the main text nor in the methods. It seems that the WGCNA function to merge modules based on the similarity of the eigengenes was not used (this function is implemented in the R package). Therefore, the modules look rather fractured. Why are, for example, the modules brown, turquoise and orange merged to module A, when they do not seem to be close to each other in Figure S2a? Also, a numbering of the colors would be of advantage, since "brown" and "orange" are not very clearly defined. In Figure S2a, there would be at least 2 candidates for "orange" and 2 for "brown". Also, all those colors would be very hard to separate visually for a color-blind person. Therefore, some additional labeling of the modules would be great. The WGCNA R package offers a custom figure of a dendrogram of module eigengenes and a heatmap. This can be easily labeled with color-names and/or cluster numbers.
- Survival analyses and correlations were performed with R. Why were protein data not analyzed with R after pre-processing and normalization in Perseus? Differential expression could be analyzed with limma package and the WGCNA could be done with WGCNA R package. Both packages have very good tutorials. This would give you the option to have better control on the analyses. I do not think that you should repeat all your analyses, but I think you should consider these options for future analyses.
- WGCNA was developed by Langsteiner and Horvath and should be properly cited.
- Methods: Separate LC-MS/MS data analysis and additional analyses (WGCNA, survival analyses) in the methods section. The overexpression analysis of WGCNA clusters (Figure 3a) is not mentioned in the methods.
- The survival analyses should be a separate section in the methods. The Kaplan-Meier plots should show the number of patients at risk in each group (e.g. include a risk table in the plot). How groups were separated should also clearly be stated in the methods section. It is mentioned for the proteomics data in the main text, but not for the TCGA data. Also, association of PYCR1 with clinical parameters could be tested in a multivariate model (are PYCR1 levels in the post-treatment group still significant if adjusted to e.g. tumor grade at diagnosis, tumor size at diagnosis and Miller & Payne score?). For the multivariate model (Figure S5d), the number of events should be stated, as they limit the number of predictors that can be included in the model. The abbreviations OS and RFS are not introduced (I presume it means overall survival and recurrence free survival?).
- Can the impact of PYCR1 also be shown in other breast cancer datasets (protein or RNA)?
- Figure 5a+b and 5 d+e: no statistical tests were used to assess differences between groups. How can it be "significantly reduced", if no test was applied?
- Figure 5f and Figure 6b: Three groups should not be compared with 2 t-tests, but with the appropriate omnibus test and post-hoc tests.
- Statistical methods used for group comparison should be properly stated in the manuscript: e.g. test for normality and homogeneity of variances used and then which test was chosen. T-tests should not be used for multiple comparisons of groups.

Minor

- Page 6: It is not clear what the ordinal groups were for the spearman rank correlations (binning of RFS?)
- Page 7: It is not entirely clear how the global pattern analysis was conducted. Were the students t-tests performed between each of the three groups pre-, post-treatment and adjacent normal tissue for each protein?

- Page 10: Specific reason for selecting PYCR1, ALDH18A1? (Based on Figure 4b, there are others with better or similar correlation)
- Page 10: Any significances in figure 4d?
- Page 43: Number of included patients not clearly explained for cox hazard ratio. Are all patients included?
- Figures: Often missing significance indicator e.g. Figure 5b
- Figure 6e: PYCR1 gRNA2 lower viability than PYCR1 gRNA1; however, significance lower (no gRNA)?
- Supplementary Figure 5 exists twice
- It would be possible to use t-SNE instead of PCA, however, as clustering was performed afterwards, PCA is fine too
- Figure 2e - p-values need to be adjusted for multiple testing
- Figure 3a: WGCNA: module-trait association: p-values need to be adjusted
- WGCNA: association of modules to biological processes: which gene sets were used? It is mentioned in figure legend, but should also be mentioned in the text
- The metabolic flux experiment was not discussed properly in the methods and figure legends of figure 5c are not labeled properly. What does M+0-6 indicate? It is not stated anywhere in the text, figure legends or methods.
- Figure 2c is red-green - change to color-blind friendly colors

Reviewer #2:

This study is a relatively small (35 patients) but carefully crafted study that uses proteomics to find patterns associated with response to breast cancer neoadjuvant treatment. The authors managed to obtain good proteome coverage and also ended up nearly 4,000 proteins that were detected in every sample and that formed the basis for further analysis. The authors performed a series of interesting types of data analysis and picked out a few proteins for follow up studies. The work is interesting and is a candidate for publication in MSB. However, the presentation of the work needs improving as outlined in a few places below.

- Introduction: the results of the breast cancer CPTAC study should at least be mentioned, better used to compare to the current study. the reviewer understands that the current study focusses on treatment response but comparison to existing proteomic data may lend further support to or deprioritize some of the stated hypothesis generated in this study.
- Figure 1c: p-values for both comparisons should be shown.
- The following sentence of the results section is perhaps an over-interpretation. "The pre-treatment samples and tumor adjacent normal samples showed a higher average correlation (R=0.51 and R=0.50, respectively) compared to post-treatment samples (R=0.46)." And what do the authors want to imply by this statement.
- The text associated with Figures 2a and 2b are a bit hard to follow as no measures of statistical significance are shown in the figure. 2b is just a heat map. Please clarify
- Figure 2c: why pick out these proteins in particular? They all come from the dominant pattern. Any follow up on these to support their role in creating the pattern and/or involvement in therapy response?
- Figure 2e: an interesting analysis but I wonder a) why a cutoff of 1,5 was chosen for this data

exploration? What happens if other cutoffs are chosen, can one 'optimize' the analysis in a way that increases correlation and statistical significance? Also, the correlations are mostly quite weak despite reaching statistical significance. The predictive value of the pattern for a given patient is therefore likely very low / non-existent. The authors should make this clear in the text.

- Similarly, the WGCNA analysis is useful but the authors should clearly point out that the correlations they observe, albeit statistically significant, may not have any predictive power. Can the authors add text on the effect sizes? Statistical significance without strong effect sizes often no practical value.
- It does not become very clear what one sees in Figure 3b or if this is important. Consider moving to supplement
- Figure 4a: again the text and figure do not follow very well. I cannot see which module is the one with Pearson correlation 0.46, $p=0.005$. Also, it is not really clear what is shown on the x-axis. It says 'module eigengene'. Does each of the histogram bars represent a patient? If not, please use less abstract language to guide readers in what they see/read (I am likely already lost here...)
- Figure 4b: again, it completely unclear what is on the axis. The text does not help to discern this and no reference is made to the methods section where this may be explained. If 4b shows proteins from one of the histogram bars in 4a, this could be made clear in the figure.
- Even though the authors say that "PYCR1 and ALDH18A1, were among the top ten proteins whose expression pattern positively correlated with the yellowgreen eigengene module and with relapse", they a) do not refer to where this list may be found and b) why they picked these ones from that group. It is not visually clear from Fig 4b why one would focus on these proteins over any other one. It is fine if these examples were picked after e.g. reading around the literature or other information that made these proteins interesting but please explain.
- Related to the above, analysis of the proteins in that pathway showed the same trends in all patients. While this is interesting, it is a bit off the topic of the paper that sought to identify factors informative to understand neoadjuvant therapy.
- The following statement is not well justified: "High expression of other proline biosynthesis proteins in post-treatment residual tumors was not associated with survival (Cox proportional hazard multivariate RFS $p = 0.07$, Supplementary Figure. 5d)." The p value of 0.07 is just outside what the authors consider significant. Given that p -values are often used as yes/no categories, they really are not. Actually, one might argue that a p -value of 0.05 makes them more likely to behave similar to the other members of the pathway.
- This reviewer is not sure what the Crisp experiment really shows. Knocking out PYCR1 in any cell line would reduce secretion of proline. So this experiment does not really validate anything other than that the protein is involved in the process.
- The discussion is overly long, should contain less repetition of introduction, results and instead focus on discussing what all the results obtained may mean in the context of the overall aims of the study. Given the depth of the discussion, this reviewer could write down at least another 10 experiments the authors should perform to relate their work to the discussion. This would clearly be outside the scope of the current manuscript. Hence, one might advise to focus on the discussion of the key points of the present work.
- For example, patterns 3 and 4 are sort of unique to this study and the discussion could focus more on these. Another example, the discussion on the WGCNA discusses many proteins that do not feature anywhere in the results so it is not clear why readers have to read about them. If other studies deem these proteins important, the authors should focus on if/how the proteomics data is in line with these prior studies.
- Perhaps most importantly, I am missing a discussion about what the results of the discussion may mean for what is mentioned in the title - "response to breast cancer neoadjuvant treatment". The authors may also want to discuss how/if their results may help/change diagnosis or clinical management. This may well be a future vision but would be of great guidance for readers.

Reviewer #3:

Shenoy et al. investigated whether protein expression patterns in primary breast cancer tissues are predictive for chemotherapy responsiveness. The authors have chosen an unbiased proteomic profiling approach (LC-MS/MS proteomic analysis of FFPE tissues) and analyzed matched tumors before and after neoadjuvant chemotherapy treatment with patient-matched tumor adjacent normal tissues from 35 breast cancer patients. After in-depth analyses of the proteomic data, the authors have defined eight protein expression patterns of which two correlated with a better chemotherapy response. Furthermore, the authors identified 2 proteins, PYCR1 and ALDH18A1, whose expression correlated with a high risk of tumor relapse in breast cancer patients. The metabolic function of both proteins within the TCA cycle have been validated as well as a pro-invasive function of PYCR1 in the human breast cancer cell line MCF7 in vitro. But the authors fall short in proving that high levels of PYCR1 promote relapse.

Major points

- The manuscript is well written and has a logical and understandable structure. The figures are well presented.
- It is unclear how the authors differentiate between the terms protein patterns, protein network, protein modules and signaling pathways.
- The authors have chosen an unbiased protein expression analysis approach on 35 primary breast cancer tissues, each matching pre and post chemotherapy treatment. As pointed out, such an analysis is novel and very interesting for the research field. However, the findings on PYCR1 are not novel. High gene expression of PYCR1 have been found in different types of cancer as well as its expression correlation with a worse chemotherapy response.
- The functional analyses on PYCR1 is kept rather short and is not convincing. The authors should test the function of PYCR1 in pathophysiologically relevant in vitro models and in a pre-clinical in vivo mouse model. First, the authors used the CRISPR/Cas9 technology to knockout PYCR1 in MCF7 cells only. Why just analyzing the role of PYCR1 in one cell line only? Secondly, why using a migration assay to prove the function of the protein in the process of tumor relapse? Thirdly, treating knockout cells with the clinically used chemotherapies (Doxorubicin, Paclitaxel and Cyclophosphamide) is relevant, but it should be described why the indicated concentrations have been used. The authors should perform a spheroid assay with drug holiday and measurement of the colony forming capacity. The necessary experiment however would be an in vivo relapse model.
- Figure 2b) Most of the proteins fall in pattern 3 (anti-correlative for relapse-free survival (RFS)), followed by pattern 4. Both patterns imply that there is no difference in protein expression comparing pre- vs post-treatment, which means those expression patterns are in the tumor cells from the beginning and are independent of chemotherapy treatment. This is a very interesting observation that should be discussed. In general, the authors performed a very comprehensive analyses on the proteomic data, however, the biological meaning of the findings should be more highlighted/discussed.

Minor points

- The term "protein expression" is not accurate, since per definition only genes are expressed. The authors should use " protein abundance or protein level" instead.

-
- Figure 3a) Eigengene module "yellowgreen" has the opposite pattern to "Paleturquoise. How does PYCR1 expression changes? Does it behave inversely?
- Figure 4g) What is the difference between this graph and the graph presented in supplementary Fig 5b?
- Figure 6d) No biological replicates in the proliferation assay (KO PYCR1)
- How does the cells compensate for a loss of PYCR1 since there is no difference in cell proliferation?
- Page 17 bottom: "Reference source not found"?

Point by Point response to Reviewer's comments

We thank the reviewers for their valuable comments on our manuscript "Proteomic patterns associated with response to breast cancer neoadjuvant treatment". In the revised manuscript, we address these points in the following ways.

We reinforced the WGCNA findings using supervised statistical test with multiple hypotheses testing correction. There is a large overlap in these two analyses and most importantly, we show that proteins remain unaltered post treatment in patients with poor prognosis and PYCR1 is associated with both drug response and relapse.

Next, we have extended some of the PYCR1 functional assays to MDA-MB-231 triple negative breast cancer cell line.

Finally, we validated the effect of PYCR1 suppression on drug response in-vivo using orthotopically injected breast cancer cell line model.

Overall, we added 21 figure panels; 5 to the main figures, 10 to Extended view and 6 in the appendix figures.

Reviewer#1:

The manuscript "Proteomic patterns associated with response to breast cancer neoadjuvant treatment" is interesting and presented with good language and a nice flow. The proteomic data and the comparison of the three tissue types (adjacent normal, pre-treatment, post-treatment) are well done. However, Shenoy et al. should be more thorough with the statistical tests they use. Often, p-value adjustment was not done after multiple testing, especially for the WGCNA analyses. Multiple groups should not be compared with multiple t-tests. Also, the statistics should be presented more thoroughly in the methods section, especially the survival analyses. In addition, the most important finding of the authors, namely the important role of PYCR1, is not validated by in vivo experiments in breast cancer mouse models to support their findings and supplement their proteomics bioinformatics hypothesis.

We thank the reviewer for his/her interest our manuscript, and appreciate the valuable and constructive comments. You can find the detailed answers below.

Major

- The possible role of PYCR1 loss of function for breast cancer tumor growth should be validated in an in vivo setting, especially in combination with the chemotherapeutics used in fig. 6e-i

We thank the reviewer for this comment. Following this suggestion we performed the in-vivo experiments, and we believe that these results dramatically strengthened the study, as described below. The in-vivo effect of PYCR1 loss of function in triple negative breast cancer has already been shown (Loayza-Puch *et al*, 2016), however, the effect on drug response, as well as the effect

on luminal breast cancer have not been studied before. Following the reviewer's request, we analyzed the growth of PYCR1 KO cells in-vivo. We examined this effect in both MCF7 ER+ cell line (in-which we performed most of our experiments) as well as MDA-MB-231 triple negative cell line. In-vivo analysis of MCF7 growth showed a very significant reduction in tumor burden upon treatment with paclitaxel and doxorubicin only in the KO cells and not the WT MCF7 cells (Reviewer figure 1). The figure panels are inserted in Main figure 6, EV Figure 4 and Appendix figure S6. Since MDA-MB-231 tumors did not achieve the desired size to start treatment, we only show tumor growth in this system. These results show the importance of PYCR1 to the growth of both ER+ and TNBC, however the functional significance may differ. While TN cells depend on PYCR1 for growth, migration and invasion; growth of ER+ depends on PYCR1 mainly in the context of drug response. We associate these differences with the metabolic differences between subtypes, as discussed in the revised manuscript.

Reviewer Figure 1: Effect of PYCR1 KO in-vivo

A) Tumor volume measurements for 26 days in MCF7 injected NSG mice. CRISPR control and PYCR1 KO tumors with ($n=8$) and without treatment ($n=5$) are shown. Data represents mean \pm SE. Pax: Paclitaxel, Dox: Doxorubicin. **B)** Bar plot indicates mean \pm SE of tumor volume measurements for day 26. Groups are compared by one-way ANOVA followed by Tukey's multiple comparisons test for pairwise group comparisons. Corrected p -values are reported as follows * $p < 0.05$, ** $p < 0.01$, *** $p < 0.001$. **C)** Tumor volume measurements for 26 days in MDA-MB-231 injected NSG mice. CRISPR control ($n=6$) and PYCR1 KO tumors ($n=6$) without treatment are shown. Data represents mean \pm SE. **D)** Bar plot indicates mean \pm SE of tumor weight measurements for day 26. Samples are compared using paired student's t -test. p -values are reported as follows * $p < 0.05$, ** $p < 0.01$, *** $p < 0.001$, **** $p < 0.0001$.

- Figure 4b; WGCNA: What are the adjusted p-values for gene significance (GS) and module membership (MM) in the yellowgreen module for PYCR1 and ALDH18A1, respectively? Was an adjusted-p-value cutoff used to select only proteins with a significant GS and MM?

We appreciate the reviewer's comment and tried to address it by adjusting the p-values for gene significance (GS) and module membership (MM). Adjusted p-values are 0.5 (GS), 0.008 (MM) for PYCR1 and 0.6 (GS), 0.02 (MM) for ALDH18A1. Since the GS was insignificant upon FDR control, we used a complementary approach to examine association with relapse and response (described below). Furthermore, we used an adjusted p value cutoff of module membership for filtering the protein lists. PYCR1 and ALDH18A1, were among the top 10 proteins in the module with lowest GS and MM p values (Figure 4b, Dataset EV5D). The significance of the WGCNA results is toned down in the revised manuscript.

To support our hypothesis that the proline biosynthesis is associated with relapse and response, we asked whether differentially expressed proteins in better responders to treatment were also altered in non-responders (Reviewer figure 2, and main text Figure 3). PYCR1 was significantly downregulated upon treatment in responders (follows pattern1) and was unaltered in non-responders to treatment (follows pattern 3; paired t-test, FDR 5 %). PYCR1 levels also remained unchanged upon treatment in patients with relapse when compared to patients with no relapse. Altogether, these results reinforce the association of PYCR1 with drug response and tumor relapse.

Reviewer Figure 2: Supervised analysis.

A) Volcano plot shows significantly altered proteins in better responders before and after treatment. Samples are compared by paired student's t-test, FDR 5%. These proteins are not significantly altered in Poor responders. **B)** Same as (a) but comparing patients with and without relapse. PYCR1 is highlighted.

- WGCNA : On what basis were the modules merged? It is not stated in the main text nor in the methods. It seems that the WGCNA function to merge modules based on the similarity of the

eigengenes was not used (this function is implemented in the R package). Therefore, the modules look rather fractured. Why are, for example, the modules brown, turquoise and orange merged to module A, when they do not seem to be close to each other in Figure S2a? Also, a numbering of the colors would be of advantage, since "brown" and "orange" are not very clearly defined. In Figure S2a, there would be at least 2 candidates for "orange" and 2 for "brown". Also, all those colors would be very hard to separate visually for a color-blind person. Therefore, some additional labeling of the modules would be great. The WGCNA R package offers a custom figure of a dendrogram of module eigengenes and a heatmap. This can be easily labeled with color-names and/or cluster numbers.

We apologize for the lack of clarity. The modules were only combined to analyze the collective enrichment of biological processes associated with clinical features. No module eigengene was calculated for the merged module and used for further analysis. To clarify this and differentiate between the WGCNA modules and the merged clusters, we call it 'Protein cluster A' instead of module A. The dendrogram of module eigengenes and heatmap with module color names have been included (Reviewer figure 3, Appendix Figure S2)

Reviewer Figure 3: Module eigengene heatmap and dendrogram.

Heatmap of eigengene similarity matrix and associated dendrogram. Modules are indicated by module colors.

- Survival analyses and correlations were performed with R. Why were protein data not analyzed with R after pre-processing and normalization in Perseus? Differential expression could be analyzed with limma package and the WGCNA could be done with WGCNA R package. Both packages have very good tutorials. This would give you the option to have better control on the analyses. I do not think that you should repeat all your analyses, but I think you should consider these options for future analyses.

We thank the reviewer for these suggestions. We reanalyzed the WGCNA data in R, and we see that the main conclusions surrounding PYCR1 or treatment response do not change or improve upon adjusting the different parameters such as 'deepSplit', 'mergeCutHeight', 'moduleMergeUsingKME' and 'mergeCloseModules'. Using the R package, we have now generated the requested figures and included them in the revised version of the manuscript. See reviewer figure 3.

- WGCNA was developed by Langsteiner and Horvath and should be properly cited.

We apologize for missing important citations. We have included the following references in the revised version.

Langfelder P & Horvath S (2008) WGCNA : an R package for weighted correlation network analysis. BMC Bioinformatics.

Zhang B and Horvath S (2005) A General Framework for Weighted Gene Co-Expression Network Analysis, Statistical Applications in Genetics and Molecular Biology.

Storey JD, Taylor JE, and Siegmund D. (2004) Strong control, conservative point estimation, and simultaneous conservative consistency of false discovery rates: A unified approach. Journal of the Royal Statistical Society.

- Methods: Separate LC-MS/MS data analysis and additional analyses (WGCNA, survival analyses) in the methods section. The overexpression analysis of WGCNA clusters (Figure 3a) is not mentioned in the methods.

Following the reviewer's request, LC-MS/MS data analysis has been separated from the rest of the bioinformatics and statistical analyses. Missing details have been added to the methods and figure legend.

- The survival analyses should be a separate section in the methods. The Kaplan-Meier plots should show the number of patients at risk in each group (e.g. include a risk table in the plot). How groups were separated should also clearly be stated in the methods section. It is mentioned for the proteomics data in the main text, but not for the TCGA data.

Risk table has been added to all the survival plots. For survival analysis using TCGA data, we used the cBioportal database. High or low mRNA expression was determined by the number of standard deviations (SD) from the mean (mRNA Expression z-Scores of RNA Seq V2 RSEM values). Survival analysis is separated in the methods and patient stratification details for all survival analyses have been added to the methods section.

- Also, association of PYCR1 with clinical parameters could be tested in a multivariate model (are PYCR1 levels in the post-treatment group still significant if adjusted to e.g. tumor grade at diagnosis, tumor size at diagnosis and Miller & Payne score?).

We thank the reviewer for this comment. Following the reviewer's request, PYCR1 abundance levels were corrected for confounding variables such as tumor grade, tumor size and Miller and Payne response score using 'Limma' package in R, and checked for association to survival. New graphs indicate that post treatment PYCR1 levels are still significantly associated with survival when corrected for tumor size and grade (Reviewer figure 4A). In addition, we included all three variables along with PYCR1 in a multivariate model. Global p value remains significant at 0.002 (Reviewer figure 4B, Appendix Fig S4D).

Reviewer Figure 4: Kaplan Meier plots for normalized data.

A) Kaplan Meier survival curve for PYCR1 levels in residual cancer in the current proteomics data. Cox univariate p value and hazard ratio with 95% CI indicated. PYCR1 abundance level was first normalized to Tumor grade, Tumor size using 'Limma' package in R. Protein level was defined as low /high based on the median abundance value. **B)** Same as (A) but normalized to Miller and Payne response score. **C)** All four variables were tested in a multivariate model. Number of events and global p value are indicated. All 35 patients were included in the analysis.

- For the multivariate model (Figure S5d), the number of events should be stated, as they limit the number of predictors that can be included in the model.

The number of events in multivariate analysis is 7 (equal to the number of relapses). The information is provided in the methods section and figures of the revised version.

- The abbreviations OS and RFS are not introduced (I presume it means overall survival and recurrence free survival?).

We apologize for this mistake. OS and RFS refer to overall survival and recurrence free survival. We have introduced these variables in the results and legends.

- Can the impact of PYCR1 also be shown in other breast cancer datasets (protein or RNA)?

In the previous version of the manuscript we showed the impact of PYCR1 RNA expression and disease-free survival in the invasive breast cancer samples from TCGA cohort (Appendix Figure S4G). To address the reviewer's comment we mined the recently published proteomics dataset comprising paired treated and untreated samples from a small neoadjuvant chemotherapy cohort (Satpathy *et al*, 2020). We compared PYCR1 protein levels between paired tumor samples taken before and after 72 hours of chemotherapy. While complete responders to treatment show a significant reduction in PYCR1 levels following chemotherapy, Non-Responders show no significant change (Reviewer Figure 5, Appendix Figure S4H).

Reviewer Figure 5: PYCR1 in other proteomics dataset

Log₂ Normalized PYCR1 level (TMT reporter ion intensity) differences between matched Pre and On treatment samples (72 hrs after initiation of neoadjuvant chemotherapy). Patients are separated based on the overall response to Neoadjuvant chemotherapy. Paired samples are compared using Wilcoxon-rank sum test.

- Figure 5a+b and 5 d+e: no statistical tests were used to assess differences between groups. How can it be "significantly reduced", if no test was applied?

We used Kruskal Wallis test followed by Dunnet's test to compare the WT cells to the two different KO cells. P values have been included and the statistical tests are indicated in the text.

- Figure 5f and Figure 6b: Three groups should not be compared with 2 t-tests, but with the appropriate omnibus test and post-hoc tests.

In figure 5 and 6b we compare the CRISPR control to either of the PYCR1 KO samples and not between the two KO samples. We performed Kruskal Wallis test followed by Dunnet's test with Benjamini-Hochberg (BH) FDR correction to compare a single control group to the two KOs. Corrected p values are indicated.

- Statistical methods used for group comparison should be properly stated in the manuscript: e.g. test for normality and homogeneity of variances used and then which test was chosen. T-tests should not be used for multiple comparisons of groups.

We used Shapiro Wilk test for normality and Bartlett test for homogeneity of variances. Wherever applicable, we used non-parametric tests. Similarly, while comparing multiple groups we correct for multiple hypothesis testing and report corrected p values. The details are added in the figure legends.

Minor

- Page 6: It is not clear what the ordinal groups were for the spearman rank correlations (binning of RFS?)

We first calculated sample-sample correlation for all samples in the dataset. Next, we calculated the average Spearman Rank correlation within each group (Normal, Pre-treatment, Post treatment) and between groups (Normal: Pre-treatment, Normal: Post-treatment, Pre-treatment:Post-treatment). Samples were not grouped based on binning of RFS.

- Page 7: It is not entirely clear how the global pattern analysis was conducted. Were the students t-tests performed between each of the three groups pre-, post-treatment and adjacent normal tissue for each protein?

We apologize for the lack of clarity. The global pattern analysis was conducted by performing 3 Student's t-tests (paired) between each of the three groups for each protein with FDR correction.

- Page 10: Specific reason for selecting PYCR1, ALDH18A1? (Based on Figure 4b, there are others with better or similar correlation)

We performed supervised analysis on the data and looked for proteins associated with both treatment response and relapse. PYCR1 and ALDH18A1 are significantly associated with both clinical features. Since their protein abundance levels in residual tumors is also associated with clinical outcome (WGCNA analysis) we chose these proteins for further validation. In addition,

identification of several members of one metabolic pathway is rather unique and suggests that the pathway has functional importance.

- Page 10: Any significances in figure 4d?

Corrected p-value indicator has been added to the figure.

- Page 43: Number of included patients not clearly explained for cox hazard ratio. Are all patients included?

All 35 patients were included in the analysis. The missing information has now been added to the methods and figure.

- Figures: Often missing significance indicator e.g. Figure 5b

Missing significance indicator was added.

- Figure 6e: PYCR1 gRNA2 lower viability than PYCR1 gRNA1; however, significance lower (no gRNA)?

We apologize for this typo. Indeed, PYCR1 gRNA2 has lower viability and lower p value.

- Supplementary Figure 5 exists twice

Repeated figure was removed.

- It would be possible to use t-SNE instead of PCA, however, as clustering was performed afterwards, PCA is fine too

We thank the reviewer for the advice. We kept the PCA in the revised version.

- Figure 2e - p-values need to be adjusted for multiple testing

Adjusted p values have been indicated.

- Figure 3a: WGCNA: module-trait association: p-values need to be adjusted

Upon the reviewer's request, we calculated adjusted p-values for module trait associations. However, corrected p values do not pass the significance cutoff for any comparison. Adjusting the WGCNA parameters using the R package does not improve the results. Given the limited significance of the WGCNA, we shortened the description of these results and moved some of the figure panels to the supplementary figures, and further strengthened and supported our main conclusions using supervised analysis (with FDR correction), which shows a large overlap with the WGCNA (Figure 3A & Figure EV1).

- WGCNA: association of modules to biological processes: which gene sets were used? It is mentioned in figure legend, but should also be mentioned in the text.

For unsupervised WGCNA analysis, gene annotations including GOBP, GOMF, GOCC, GSEA gene sets from MSigDB and KEGG pathway were added from Uniprot, and Fisher Exact test was performed with the entire identified data of 7600 proteins as background. The missing details have been added to the Methods section.

- The metabolic flux experiment was not discussed properly in the methods and figure legends of figure 5c are not labeled properly. What does M+0-6 indicate? It is not stated anywhere in the text, figure legends or methods.

A metabolite with n carbon atoms can have 0 to n of its carbon atoms labeled with ^{13}C , resulting in isotopologues that increase in mass (M) from M+0 (all carbons unlabeled e.g. ^{12}C) to M+n (all carbons labeled e.g. $^{13}\text{C}_n$). To clarify this, the missing details have been added to the legends and method sections.

- Figure 2c is red-green - change to color-blind friendly colors

Figure 2C has been changed to color-blind friendly colors.

Reviewer #2:

This study is a relatively small (35 patients) but carefully crafted study that uses proteomics to find patterns associated with response to breast cancer neoadjuvant treatment. The authors managed to obtain good proteome coverage and also ended up nearly 4,000 proteins that were detected in every sample and that formed the basis for further analysis. The authors performed a series of interesting types of data analysis and picked out a few proteins for follow up studies. The work is interesting and is a candidate for publication in MSB. However, the presentation of the work needs improving as outlined in a few places below.

We thank the reviewer for the interest in our work. We addressed all of the reviewer's comments as detailed below.

- Introduction: the results of the breast cancer CPTAC study should at least be mentioned, better used to compare to the current study. the reviewer understands that the current study focusses on treatment response but comparison to existing proteomic data may lend further support to or deprioritize some of the stated hypothesis generated in this study.

Missing references have been added.

- Figure 1c: p-values for both comparisons should be shown.

Significant P values for pairwise comparisons have been added to the figure.

- The following sentence of the results section is perhaps an over-interpretation. "The pre-treatment samples and tumor adjacent normal samples showed a higher average correlation ($R=0.51$ and $R=0.50$, respectively) compared to post-treatment samples ($R=0.46$)." And what do the authors want to imply by this statement.

While the correlation differences are small, we wanted to highlight that varied response to treatment in patients is reflected in the comparatively lower correlation of post treatment samples. Nevertheless, following the reviewer's comment, we rephrased this section to "*Average correlation between matched pre- and post-treatment samples was 0.58, and the correlations with matched tumor adjacent normal was markedly lower (post-treatment: tumor adjacent normal ($R=0.43$) and pre-treatment: tumor adjacent normal ($R=0.39$)).*"

- The text associated with Figures 2a and 2b are a bit hard to follow as no measures of statistical significance are shown in the figure. 2b is just a heat map. Please clarify.

Figure 2B is a heatmap of proteins that pass the global pattern analysis test (patterns across all samples with FDR correction). The heatmap shows that Pattern 3 proteins are dominant in this dataset. In the revised version we moved Figure 2B to Appendix Figure S1.

- Figure 2c: why pick out these proteins in particular? They all come from the dominant pattern. Any follow up on these to support their role in creating the pattern and/or involvement in therapy response?

We thank the reviewer for this comment. We picked out AKT1, STAT1 and mTOR, as they are well-studied oncogenes. They followed Pattern 3 in majority of the patients in our cohort. Following the reviewer's comment, we performed a centrality analysis on all the 736 Pattern3 proteins and highlighted top 10% of the proteins with the highest betweenness centrality (Reviewer Figure 6, Figure 2B). AKT1, MTOR and STAT1 are among the top 10% most central proteins in the network. In the follow-up supervised and unsupervised analyses we also see that STAT1 is associated with chemotherapy response.

Reviewer Figure 6: Network analysis of Pattern 3 proteins

Pattern 3 Protein network based on global pattern analysis: Node size and color is based on betweenness centrality score of each node in the network. Top 10% of the most central nodes are indicated.

- Figure 2e: an interesting analysis but I wonder a) why a cutoff of 1,5 was chosen for this data exploration? What happens if other cutoffs are chosen, can one 'optimize' the analysis in a way that increases correlation and statistical significance? Also, the correlations are mostly quite weak despite reaching statistical significance. The predictive value of the pattern for a given patient is therefore likely very low / non-existent. The authors should make this clear in the text.

A 1.5 cut-off, as any other cutoff is random (but commonly used in the literature), reflects the trend of changes between normal, pre- and post-treatment samples. In fact, using 1.5-, 2-, or 3-fold change shows a similar ratio between the patterns and similar correlations to relapse free survival time (Reviewer Figure 7). Increasing the fold change cutoff results in fewer protein patterns per patient (Reviewer Figure 7B). Moreover, we don't refer to this analysis as a predictor of response but rather highlight the importance of pattern 3 proteins as they are associated with patient outcome.

Reviewer Figure 7: Patient-wise pattern analysis.

A) Bar plot with percentages of proteins following each pattern in 35 patients using four different fold change cutoffs. **B)** Bar plot shows the number of proteins that form a pattern when different fold change cutoffs are used. **C)** Spearman rank correlation between percent of proteins following each of the patterns with relapse free survival in each patient. Black border indicates corrected p value <0.05.

- Similarly, the WGCNA analysis is useful but the authors should clearly point out that the correlations they observe, albeit statistically significant, may not have any predictive power. Can the authors add text on the effect sizes? Statistical significance without strong effect sizes often no practical value.

Following the comments from the other reviewers we further examined the statistical significance of the WGCNA. Following these analyses, and given the limited statistical power of the WGCNA, we added complementary supervised analysis of association with response and with relapse. In the WGCNA, the correlations of module eigengene to clinical trait were modest (0.3 to 0.5) and did not pass FDR correction. We therefore selected correlations with effect size 0.3 (absolute values) and above, and examined their overlap with the complementary supervised analysis, thus strengthening the clinical trait associations of these proteins and modules.

- It does not become very clear what one sees in Figure 3b or if this is important. Consider moving to supplement

Figure 3b has been moved to the supplementary material (Figure EV3B). This heatmap shows all the proteins in the modules associated with M&P score. While the WGCNA modules were built on post treatment samples alone, when visualizing each protein with matched pretreatment and healthy samples grouped according to M&P score, one can see that the pattern followed by these proteins differ between different response groups. In support of our hypothesis, proteins associated with treatment response behave as either pattern 3 or 4 in poor responders whereas the same proteins behave as patterns 1 and 2 in better responders.

- Figure 4a: again the text and figure do not follow very well. I cannot see which module is the one with Pearson correlation 0.46, $p=0.005$. Also, it is not really clear what is shown on the x-axis. It says 'module eigengene'. Does each of the histogram bars represent a patient? If not, please use less abstract language to guide readers in what they see/read (I am likely already lost here...)

We apologize for the lack of clarity. Figure 4A is a barplot of the module eigengene expression for the yellowgreen module. Each bar represents a patient and the bars are separated between Relapse and No Relapse. Clearer details are now added to the main text and the figure legend.

- Figure 4b: again, it completely unclear what is on the axis. The text does not help to discern this and no reference is made to the methods section where this may be explained. If 4b shows proteins from one of the histogram bars in 4a, this could be made clear in the figure.

We apologize for missing these explanations. Figure 4B shows the scatter plot of the proteins in the yellowgreen module in association with the clinical feature. The y axis represents gene significance (GS), namely the correlation between each protein in the module to the clinical trait (i.e. relapse). X axis represents the module membership (MM) which is the correlation of each protein in the module to the module eigengene. We have expanded the terms and added explanations in the methods section.

- Even though the authors say that "PYCR1 and ALDH18A1, were among the top ten proteins whose expression pattern positively correlated with the yellowgreen eigengene module and with relapse", they a) do not refer to where this list may be found and b) why they picked these ones from that group. It is not visually clear from Fig 4b why one would focus on these proteins over any other one. It is fine if these examples were picked after e.g. reading around the literature or other information that made these proteins interesting but please explain.

The list of all proteins in the yellowgreen module are indicated in Figure 4B and in expanded view Dataset (EV5), which includes the module membership and gene significance p values. Following the trendline one can see that PYCR1 has high module membership and gene significance (among the top 10). We selected to follow up on these proteins since they appeared in multiple analyses:

- PYCR1 and ALDH18A1 were associated with patient survival in the WGCNA analysis.
- PYCR1 and ALDH18A1 followed pattern 3 in the majority of the patients in our cohort (Fig 4D).

c. Centrality analysis of pattern 3 proteins showed that PYCR1 and ALDH18A1 are among the top 10 % of central proteins in the network (Fig 2B).

d. PYCR1 and ALDH18A1 are associated with drug response and relapse in the supervised analysis. Namely, they are significantly down regulated upon treatment in better responders but not in poor responders to chemotherapy (Fig EV1B).

e. PYCR1 and ALDH18A1 function in the same metabolic pathway, suggesting a functional role of the pathway in the cancerous phenotype.

- Related to the above, analysis of the proteins in that pathway showed the same trends in all patients. While this is interesting, it is a bit off the topic of the paper that sought to identify factors informative to understand neoadjuvant therapy.

In the initial submission, we used pattern analysis and showed that pattern 3 proteins which remain unaltered upon treatment in the majority of the patients, are associated with poor outcome. Using WGCNA analysis we were able to show that proteins of modules associated with M&P score behave as different patterns when patients are separated according to treatment response (Figure EV3B). In the revised manuscript, we use an additional supervised analysis to support the WGCNA results. In this analysis, we asked whether differentially expressed proteins in better responders to treatment were also altered in non-responders (Reviewer figure 8, and main text Figure 3). Proteins including PYCR1, were significantly downregulated upon treatment in responders (follows pattern1) and were unaltered in non-responders to treatment (follows pattern 3; paired t-test, FDR 5 %). Similarly, proteins that are upregulated in responders (pattern 2) remain unaltered in non-responders, thus identifying several proteins involved in neoadjuvant therapy response and strengthening our hypothesis that Patterns 3 and 4 are associated with poor outcome.

The manuscript includes multiple analyses, which aim to identify regulators of neoadjuvant treatment response. We aimed to cross these analyses to find the most robust factors, which could be further targeted to improve treatment response. Identification of multiple factors from the same metabolic pathway is quite unique and the repeated identification across multiple analyses reinforces the significance of this pathway. However, while the current study emphasized the proline biosynthesis pathway, we see the importance of this study also as a resource to the systems biology and the cancer research community. We expect that many additional analyses can stem from this resource and identify additional cancer regulators and potential drug targets.

Reviewer Figure 8: Supervised analysis

A) Volcano plot shows significantly altered proteins in better responders before and after treatment. Samples are compared by paired student's t-test, FDR 5%. These proteins are not significantly altered in Poor responders. **B)** Same as (a) but comparing patients with and without relapse. PYCR1 is highlighted.

- The following statement is not well justified: "High expression of other proline biosynthesis proteins in post-treatment residual tumors was not associated with survival (Cox proportional hazard multivariate RFS $p = 0.07$, Supplementary Figure. 5d)." The p value of 0.07 is just outside what the authors consider significant. Given that p-values are often used as yes/no categories, they really are not. Actually, one might argue that a p-value of 0.05 makes them more likely to behave similar to the other members of the pathway.

We agree with the reviewer and have modified the text to "In a multivariate model, proline biosynthesis pathway containing all four proline biosynthesis proteins was associated with survival, and PYCR1 was the most significant among them (Cox proportional hazard multivariate RFS, PYCR1 $p = 0.037$, global $p = 0.07$)".

- This reviewer is not sure what the Crisp experiment really shows. Knocking out PYCR1 in any cell line would reduce secretion of proline. So this experiment does not really validate anything other than that the protein is involved in the process.

The proline secretion experiments were only intended to validate the functionality of the knockout, and not to examine the effect on the tumorigenic phenotype. In addition, while PYCR1 is known to be the dominant proline biosynthesis gene we wanted to ensure that there is no compensation by PYCR2 or PYCR3 in these KO cells that would contribute to proline production. It was necessary to perform this experiment and ensure that the metabolite levels are as expected in order to perform the metabolomics experiments.

The discussion is overly long, should contain less repetition of introduction, results and instead

focus on discussing what all the results obtained may mean in the context of the overall aims of the study. Given the depth of the discussion, this reviewer could write down at least another 10 experiment the authors should perform to relate their work to the discussion. This would clearly be outside the scope of the current manuscript. Hence, one might advise to focus on the discussion of the key points of the present work.

For example, patterns 3 and 4 are sort of unique to this study and the discussion could focus more on these. Another example, the discussion on the WGCNA discusses many proteins that do not feature anywhere in the results so it is not clear why readers have to read about them. If other studies deem these proteins important, the authors should focus on if/how the proteomics data is in line with these prior studies.

Perhaps most importantly, I am missing a discussion about what the results of the discussion may mean for what is mentioned in the title - "response to breast cancer neoadjuvant treatment". The authors may also want to discuss how/if their results may help/change diagnosis or clinical management. This may well be a future vision but would be of great guidance for readers.

We thank the reviewer for these insightful suggestions. The discussion has been streamlined and shortened accordingly to focus on pattern 3 and clinical significance of our findings.

Reviewer #3:

Shenoy et al. investigated whether protein expression patterns in primary breast cancer tissues are predictive for chemotherapy responsiveness. The authors have chosen an unbiased proteomic profiling approach (LC-MS/MS proteomic analysis of FFPE tissues) and analyzed matched tumors before and after neoadjuvant chemotherapy treatment with patient-matched tumor adjacent normal tissues from 35 breast cancer patients. After in-depth analyses of the proteomic data, the authors have defined eight protein expression patterns of which two correlated with a better chemotherapy response. Furthermore, the authors identified 2 proteins, PYCR1 and ALDH18A1, whose expression correlated with a high risk of tumor relapse in breast cancer patients. The metabolic function of both proteins within the TCA cycle have been validated as well as a pro-invasive function of PYCR1 in the human breast cancer cell line MCF7 in vitro. But the authors fall short in proving that high levels of PYCR1 promote relapse.

Major points

- The manuscript is well written and has a logical and understandable structure. The figures are well presented.

- It is unclear how the authors differentiate between the terms protein patterns, protein network, protein modules and signaling pathways.

We appreciate the interest in our work. In the revised version we addressed all points raised by the reviewer, including experimental and textual questions.

- The authors have chosen an unbiased protein expression analysis approach on 35 primary breast cancer tissues, each matching pre and post chemotherapy treatment. As pointed out, such an analysis is novel and very interesting for the research field. However, the findings on PYCR1 are not novel. High gene expression of PYCR1 have been found in different types of cancer as well as its expression correlation with a worse chemotherapy response.

Our study provides a unique proteomic dataset of a matched breast cancer tumor cohort. We further follow and focus on the proline biosynthesis pathway, including biological experiments in-vitro and in-vivo (in the revised manuscript). As mentioned by the reviewer, PYCR1 has been indicated in several cancer types and has been correlated with worse chemotherapy response using mRNA expression datasets. Compared to previously published experimental results, we observe that PYCR1 KO affects tumor growth and drug response differently, and we show in-vivo effects of PYCR1 KO. In the revised version we show that PYCR1 is important for the tumorigenic phenotype of both triple negative (TN) and ER+ breast cancer, but we differentiate between the subtypes with regards to the mode of PYCR1 effect. We show that the sensitivity to chemotherapy increased upon PYCR1-KO only in MCF7 (ER+) cells, while triple negative MDA-MB-231 cells showed no significant response to treatment in-vitro and showed markedly reduced proliferation. We speculate that the metabolic shift induced by PYCR1 affects these cells differently due to their differences in metabolic functions. Lastly, in the revised manuscript we show that these results are reproduced remarkably in-vivo, where PYCR1 KO model shows significantly reduced tumor burden when treated with chemotherapeutics compared to WT tumors that show no substantial effects.

- The functional analyses on PYCR1 is kept rather short and is not convincing. The authors should test the function of PYCR1 in pathophysiologically relevant in vitro models and in a pre-clinical in vivo mouse model. The authors should perform a spheroid assay with drug holiday and measurement of the colony forming capacity. The necessary experiment however would be an in vivo relapse model.

In the revised manuscript we strengthened the PYCR1 analyses using additional bioinformatics analyses, additional experimental validations, and in-vivo functional analyses. An additional supervised analysis of the clinical proteomic data supported the association between PYCR1 abundance level and drug response and relapse. Establishing an in-vivo relapse model to study the effect of PYCR1 KO was not feasible for the current revision time frame, and especially in light of the ongoing pandemic, however we think that the in-vivo results are strong enough to prove the significance of PYCR1 in these models. Following the reviewer's request, we tested the effect of PYCR1 KO on response to paclitaxel and doxorubicin in an in-vivo breast cancer model. In order to do this, we injected the CRISPR WT and PYCR1 KO cell lines in mammary fat pad of mice and

monitored the growth of tumors for 26 days. We tested this effect of PYCR1 KO in MCF7 ER+ cell line and MDA-MB-231 triple negative cell line. Since PYCR1 KO MDA-MB-231 tumors did not achieve the desired tumor size to start treatment, we only tested the chemotherapeutics in the MCF7 injected model. We observed a very significant reduction in tumor burden upon treatment with the drugs only in the KO tumors and not the WT tumors (Reviewer figure 9). The figure panels are inserted in Main figure 6, EV Figure 4 and Appendix figure S6.

Reviewer Figure 9: Effect of PYCR1 KO in-vivo

A) Tumor volume measurements for 26 days in MCF7 injected NSG mice. CRISPR control and PYCR1 KO tumors with (n=8) and without treatment (n=5) are shown. Data represents mean \pm SE. Pax: Paclitaxel, Dox: Doxorubicin. **B)** Bar plot indicates mean \pm SE of tumor volume measurements for day 26. Groups are compared by one-way ANOVA followed by Tukey's multiple comparisons test for pairwise group comparisons. Corrected p-values are reported as follows * $p < 0.05$, ** $p < 0.01$, *** $p < 0.001$. **C)** Tumor volume measurements for 26 days in MDA-MB-231 injected NSG mice. CRISPR control (n=6) and PYCR1 KO tumors (n=6) without treatment are shown. Data represents mean \pm SE. **D)** Bar plot indicates mean \pm SE of tumor weight measurements for day 26. Samples are compared using paired student's t-test. p-values are reported as follows * $p < 0.05$, ** $p < 0.01$, *** $p < 0.001$, **** $p < 0.0001$.

First, the authors used the CRISPR/Cas9 technology to knockout PYCR1 in MCF7 cells only. Why just analyzing the role of PYCR1 in one cell line only? Secondly, why using a migration assay to prove the function of the protein in the process of tumor relapse? Thirdly, treating knockout cells with the clinically used chemotherapies (Doxorubicin, Paclitaxel and Cyclophosphamide) is relevant, but it should be described why the indicated concentrations have been used.

Following the reviewer's comment, we expanded the functional experiments in MDA-MB-231 triple negative cell line. We found that PYCR1 KO significantly reduces the proliferative capacity as well as invasion and migration capabilities of MDA-MB-231 cells. However, the KO affects chemotherapy response in MCF7 cells, but not MDA-MB-231 cells (Reviewer figure 10, Figure EV4).

The invasion and migration assays were performed in order to investigate the tumorigenic potential of the KO cells. Cancer cell invasion and migration are both markers of tumor aggressiveness and are associated with tumor relapse and metastasis. Furthermore, we hypothesized that proline biosynthesis may affect components of the extracellular matrix and thereby affect migration and invasion. Interestingly, the invasive-migratory phenotype was common to TN and ER+ cells, while proliferative and drug responses were subtype-specific.

To determine drug response concentrations, we calibrated each drug and selected the drug concentration that showed approximately 50% cell viability in either of the KO cells. We have clarified this in the Methods section.

Reviewer Figure 10: Effects of PYCR1 KO in MDA-MB-231 cell line.

A) Western blots showing PYCR1 knockout in MDA-MB-231 cells. Bar plot shows quantitative analysis of the western blot. **B)** Representative pictures of transwell migration and invasion after PYCR1 knockout. **C)** Bar plot represents mean ± SD of three biological replicates for the invasion and migration assay. **D)** Growth measurements of MDA-MB-231 wild type and PYCR1 KO cells over 96 hours. **E)** Bar plots show percentage of viable cells after treatment with 0.312 μM doxorubicin and 1.25 μM paclitaxel. Data represent mean ± SE of three biological experiments. Samples are compared using Student's t-test. Corrected p values are indicated as follows *p < 0.05, ** p < 0.01, and. *** p < 0.001.

Figure 2b) Most of the proteins fall in pattern 3 (anti-correlative for relapse-free survival (RFS)), followed by pattern 4. Both patterns imply that there is no difference in protein expression comparing pre- vs post-treatment, which means those expression patterns are in the tumor cells from the beginning and are independent of chemotherapy treatment. This is a very interesting observation that should be discussed. In general, the authors performed a very comprehensive analyses on the proteomic data, however, the biological meaning of the findings should be more highlighted/discussed.

We thank the reviewer for this comment. In the revised manuscript we asked whether differentially expressed proteins in better responders to treatment were also altered in non-responders (Reviewer figure 11, and main text Figure 3A,B). PYCR1 among others, was significantly downregulated upon treatment in responders (follows pattern1) and was unaltered in non-responders to treatment (follows pattern 3; paired t-test, FDR 5 %). PYCR1 levels also remained unchanged upon treatment in patients with relapse when compared to patients with no relapse. These results reinforce the notion that proteins (including PYCR1) that behave as pattern 3 or 4 in patients with poor prognosis and as patterns 1 or 2 in patients with good prognosis are clinically important. These findings are more thoroughly discussed in the revised version.

Reviewer Figure 11: Supervised analysis.

A) Volcano plot shows significantly altered proteins in better responders before and after treatment. Samples are compared by paired student's t-test, FDR 5%. These proteins are not significantly altered in Poor responders. **B)** Same as **(A)** but comparing patients with and without relapse. PYCR1 is highlighted

Minor points

The term "protein expression" is not accurate, since per definition only genes are expressed. The authors should use "protein abundance or protein level" instead.

We corrected the terminology throughout the text.

- Figure 3a) Eigengene module "yellowgreen" has the opposite pattern to "Paleturquoise. How does PYCR1 expression changes? Does it behave inversely?

In the initial submission we showed that PYCR1 belongs to the yellowgreen module in the WGCNA analysis, and that the yellowgreen module eigengene positively correlates to relapse. Conversely, the Paleturquoise eigengene showed a negative correlation to relapse and was associated with better treatment response. While the yellowgreen module did not show a good negative correlation to M&P score, in the revised manuscript, using supervised analysis, we show that PYCR1 is associated with both chemotherapy response as well as relapse (See Reviewer figure 11). Altogether, these results showed that the change in PYCR1 level is associated with drug response and relapse and PYCR1 levels in residual tumors is also associated with poor relapse free survival.

- Figure 4g) What is the difference between this graph and the graph presented in supplementary Fig 5b?

Figure 4G refers to overall survival and Appendix Figure S4E (revised manuscript) refers to relapse free survival.

- Figure 6d) No biological replicates in the proliferation assay (KO PYCR1)

The proliferation assay included three biological replicates. p value is indicated in the new figure. The details of technical and biological replicates is now included in the relevant figure legends.

- How does the cells compensate for a loss of PYCR1 since there is no difference in cell proliferation?

In the revised manuscript we also assessed the impact of PYCR1 KO in triple negative cell line MDA-MB-231 and monitored growth of both cell lines in-vivo. We found that PYCR1 KO significantly reduces the proliferative capacity as well as invasion and migration capabilities of MDA-MB-231 cells. However, the KO affects chemotherapy response in MCF7 cells, while having no significant impact on proliferation.

The invasive-migratory phenotype is subtype independent and common to TN and ER+ cells, while proliferation and drug response are different among the subtypes. Using Metabolomics and Seahorse analysis we showed that PYCR1 KO significantly effects central metabolism pathways such as glycolysis and TCA cycle. We therefore associate the different effects on the basal metabolic differences between TNBC and ER+ tumors (Tyanova *et al*, 2016; Yanovich *et al*, 2018). We speculate that since PYCR1 loss reduces glycolytic capacity, it affects the proliferation rate of cells that are more dependent on glycolysis for energy production (e.g. TNBC).

- Page 17 bottom: "Reference source not found"?

The references are corrected.

Thank you for sending us your revised manuscript. We have now heard back from the two reviewers who were asked to evaluate your revised study. As you will see below, they are both satisfied with the modifications made and are supportive of publication. As such, I am glad to inform you that we can soon accept your manuscript for publication, pending some minor editorial issues listed below.

Reviewer #1:

The work "Proteomic patterns associated with response to breast cancer neoadjuvant treatment" by Anjana Shenoy et al of the group of Tamar Geiger has been carefully revised, so I believe that nothing stands in the way of publication. Thanks for having me for this review of this interesting publication.

Reviewer #2:

The authors have done a good job at answering my questions. In addition, the additional data has improved the overall quality of the work quite a lot and it is nice to see that the authors included a mouse experiment.

In my view, the work is suitable for publication now.

The authors performed the requested editorial changes.

Thank you again for sending us your revised manuscript. We are now satisfied with the modifications made and I am pleased to inform you that your paper has been accepted for publication.

Corresponding Author Name: Tamar Geiger

Manuscript Number: MSB-20-9443